# SplatFormer: Point Transformer for Robust 3D Gaussian Splatting

**Yutong Chen**[1], **Marko Mihajlovic**[1], **Xiyi Chen**[2], **Yiming Wang**[1],
**Sergey Prokudin**[1,3], **Siyu Tang**[1]
ETH Zurich[1]; University of Maryland, College Park[2];
ROCS, University Hospital Balgrist, University of Zürich [3]

`sergeyprokudin.github.io/splatformer`

## Abstract

3D Gaussian Splatting (3DGS) has recently transformed photorealistic reconstruction, achieving high visual fidelity and real-time performance. However, rendering quality significantly deteriorates when test views deviate from the camera angles used during training, posing a major challenge for applications in immersive free-viewpoint rendering and navigation. In this work, we conduct a comprehensive evaluation of 3DGS and related novel view synthesis methods under *out-of-distribution (OOD) test camera scenarios*. By creating diverse test cases with synthetic and real-world datasets, we demonstrate that most existing methods, including those incorporating various regularization techniques and data-driven priors, struggle to generalize effectively to OOD views. To address this limitation, we introduce *SplatFormer*, the first point transformer model specifically designed to operate on Gaussian splats. SplatFormer takes as input an initial 3DGS set optimized under limited training views and refines it in a single forward pass, effectively removing potential artifacts in OOD test views. To our knowledge, this is the first successful application of point transformers directly on 3DGS sets, surpassing the limitations of previous multi-scene training methods, which could handle only a restricted number of input views during inference. Our model significantly improves rendering quality under extreme novel views, achieving state-of-the-art performance in these challenging scenarios and outperforming various 3DGS regularization techniques, multi-scene models tailored for sparse view synthesis, and diffusion-based frameworks.

## 1 Introduction

Novel view synthesis (NVS) focuses on transforming 2D RGB images into immersive 3D scenes, allowing users to navigate in augmented reality (AR) and virtual reality (VR) environments. Traditionally, this problem has been approached using a standard novel view interpolation protocol, where test views are sampled at fixed intervals along the trajectory of the input views. Several NVS methods have emerged based on this protocol, with 3D Gaussian splatting (3DGS) (Kerbl et al., 2023) recently gaining attention for achieving real-time and high-fidelity results in view interpolation.

However, AR and VR applications require not only smooth transitions between input views but also the ability to explore novel regions of interest from viewing angles outside the input distribution. For instance, users may want to observe a scene from high-elevation angles, often missing from captured views. While novel view interpolation has seen significant advancements, this out-of-distribution novel view synthesis (OOD-NVS) task remains under-explored, in both evaluation protocols and methodology. A related research problem is 3D reconstruction from sparse input views, where methods often hallucinate unseen content (Liu et al., 2023a; Chan et al., 2023; Kwak et al., 2024; Liu et al., 2023b). While hallucination can be beneficial for creative applications, it may be undesirable in settings that demand accurate reconstructions, such as 3D visualization of surgical procedures (Hein et al., 2024), and unnecessary in typical daily capture scenarios.

Imagine you are capturing a statue in a museum. By varying the camera's elevation and walking around the object, you might be able to capture most of its features. However, the spatial distribution of camera angles is likely uneven, even heavily skewed, creating certain out-of-distribution views

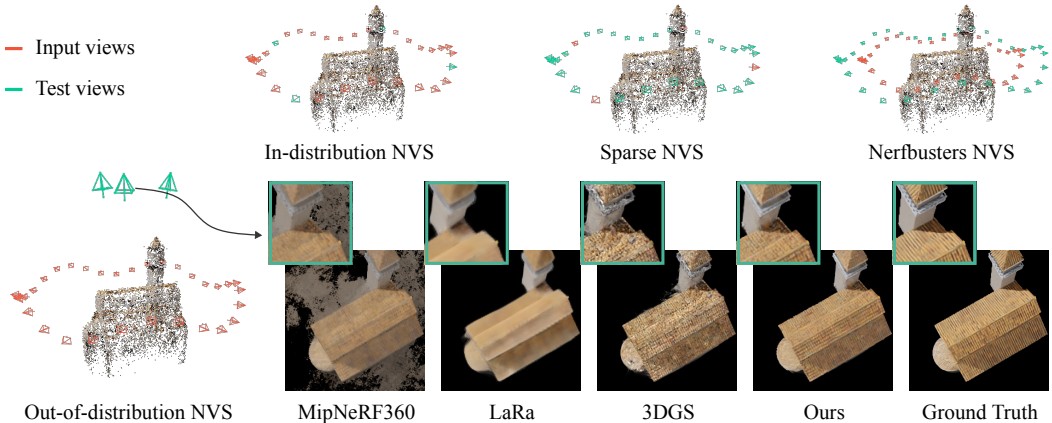

Figure 1: We investigate the out-of-distribution (OOD) novel view synthesis (NVS), where test views significantly differ from input views. This scenario contrasts with prior *in-distribution NVS*, where test views interpolate between densely captured input views, *Sparse NVS* with a few large-baseline input views, and *Nerfbusters NVS* (Warburg et al., 2023), where test views share similar angles with input views. Existing NVS methods, including MipNeRF360 (Barron et al., 2022), and those designed for sparse inputs like LaRa (Chen et al., 2024a), face challenges in this setting, while our method shows notable improvements.

where some parts of the object are only sparsely covered. An example is illustrated in Fig. 1, where input views are captured from a user's perspective, circling around an object at varying but close elevation angles. The out-of-distribution (OOD) target views observe the object from a top-down perspective, a significant deviation from the input distribution. We define this challenge as out-of-distribution novel view synthesis (OOD-NVS). We argue that this issue is practically relevant for everyday capture scenarios, yet it has been largely overlooked by the research community. To study this problem, we render 3D assets from ShapeNet (Chang et al., 2015), Objaverse 1.0 (Deitke et al., 2023), and Google Scanned Objects (Downs et al., 2022) datasets. As shown in Fig. 1, existing NVS methods perform poorly on the OOD views when restricted to low-elevation inputs, highlighting the need for a novel approach to address this problem.

Substantial research efforts have been directed towards robust 3D reconstruction with insufficient input views. First, some 3DGS variants regularize the Gaussian attributes through implicit bias in neural radiance fields (Mihajlovic et al., 2024) or geometry consistency terms (Huang et al., 2024a). Second, a number of methods attempt to exploit priors from external datasets. For example, some supervise the rendered depth maps using stereo estimators (Zhu et al., 2024), though these methods face scale ambiguity issues. Certain methods pretrain feature grids (Chen et al., 2023; Sen et al., 2023) on large datasets, but these priors are often limited to a single object category. Other methods use 2D priors from pretrained diffusion models (Sargent et al., 2024) but struggle with multi-view inconsistencies. Additionally, some feed-forward models predict 3D primitives from a few input views (Chen et al., 2024a;b; Yu et al., 2021), yet they handle no more than four images due to computational constraints, limiting their ability to leverage dense multi-view inputs. Most of these approaches are evaluated only on view interpolation or sparse-view reconstruction, failing to address the artifacts encountered in the OOD-NVS settings.

Defining an implicit regularization to improve OOD-NVS poses a significant challenge. We hypothesize that addressing this issue requires careful consideration of three key aspects: *1)* leveraging generic priors from large-scale datasets, *2)* ensuring 3D consistency in renderings, and *3)* fully utilizing the rich geometric information from all input views. To meet these needs, we propose *SplatFormer*, a novel learning-based feed-forward 3D transformer designed to operate on Gaussian splats. SplatFormer refines an initial 3DGS set—optimized using all input views—into a new, enhanced set that produces multi-view consistent 2D renderings under OOD conditions with fewer artifacts.

Our method begins by optimizing 3DGS from the input views. While this initial 3D representation effectively integrates multi-view information from the captured images, we observe that the shapes, appearances, and spatial structure of the Gaussian splats become biased toward the input view dis-

tribution. This often results in elongated Gaussian splats that cover only the thin areas projected on the input views, leading to sparse surface coverage. Furthermore, these splats can form unordered geometric structures that appear correct from the input views but exhibit significant artifacts when rendered under OOD views.

Unlike previous works that rely on hand-crafted regularization techniques (Xie et al., 2024a; Li et al., 2024b), we adapt point transformer (Zhao et al., 2021), an attention-based architecture designed for 3D scene understanding, to process the 3DGS as a point cloud set with Gaussian attributes serving as features. The attention mechanism in the point transformer learns to capture multi-view information embedded in the 3DGS, focusing on the local neighborhood within the spatial structure precomputed by the initial 3DGS. It outputs residuals that are added to the input Gaussian attributes. The updated 3DGS is then rendered from novel views, and a photometric error between the rendered and ground-truth images is minimized to train the SplatFormer. We curate large-scale training pairs of initial, flawed 3DGS sets, and ground-truth images of in-distribution and OOD views using ShapeNet and Objaverse 1.0, which are made feasible by the fast optimization of 3DGS and the availability of large-scale 3D and multi-view datasets. By training on this dataset, SplatFormer learns generalizable priors for refining 3DGS, effectively removing artifacts in the OOD views while maintaining 3D consistency.

We evaluate SplatFormer against baseline models using the proposed OOD-NVS evaluation protocols. Our experiments demonstrate that once trained, SplatFormer significantly reduces artifacts in 3DGS OOD-view renderings, showing substantial improvements in both quantitative and qualitative results for test scenes from ShapeNet and Objaverse. Additionally, we demonstrate that SplatFormer's artifact removal capabilities generalize to novel object categories in previously unseen datasets, such as Google Scanned Objects (Downs et al., 2022), as well as real-world captures. In summary, we make the following contributions:

- We introduce *OOD-NVS*, a new experimental protocol specifically designed to evaluate the performance of NVS methods when rendering 3D scenes from novel viewing angles that fall outside the distribution of input views. Our results demonstrate that existing methods struggle to generalize under the OOD-NVS protocol;
- We propose *SplatFormer*, a novel learning-based model that refines flawed 3D Gaussian splats to mitigate artifacts in OOD views. SplatFormer is the first approach to apply the point transformer to 3DGS processing, effectively leveraging multi-view information from a dense set of input views and learning a 3D rendering prior to remove artifacts;
- We demonstrate that *SplatFormer* significantly improves the performance of 3DGS-based methods on OOD-NVS tasks, achieving substantial gains in object-centric scenes, while also demonstrating potential for application in unbounded environments.

## 2 RELATED WORK

**Novel View Interpolation.** In most NVS scenarios, both input and test views are sampled from the same distribution, typically following a fixed trajectory or a hemispherical pattern, as Blender NeRF (Mildenhall et al., 2020), LLFF (Mildenhall et al., 2019), and Phototourism (Jin et al., 2021). Seminal works like NeRF (Mildenhall et al., 2020), InstantNGP (Müller et al., 2022), and 3DGS (Kerbl et al., 2023) have demonstrated strong performance under this view interpolation protocol. However, as we will demonstrate later, they encounter difficulties in rendering novel views from out-of-distribution (OOD) angles, a challenge that remains less explored.

**Sparse View Reconstruction.** Another line of research focuses on reconstructing 3D scenes from sparse set of input views, typically no more than four. Some approaches (Yu et al., 2021; Charatan et al., 2024; Wewer et al., 2024; Chen et al., 2024a;b; Fan et al., 2024b) directly predict 3D primitives from these sparse inputs. Others (Liu et al., 2023a; Sargent et al., 2024; Liu et al., 2023b; Kong et al., 2024; Yang et al., 2024; Abdal et al., 2024; Liu et al., 2024) finetune pretrained 2D diffusion models to generate novel views or repair degraded renderings. However, these methods are constrained by a limited number of input views and often rely on hallucinating unobserved regions. The sparse-view setting is well-suited for creative tasks but less applicable for accurate and faithful reconstructions when dense input views are available.

**Casually Captured Neural Radiance Fields.** Unlike the standard interpolation setup, Nerf-busters (Warburg et al., 2023) captures input and test views along separate trajectories, closely relevant to the OOD-NVS problem we are addressing. However, their input and test views remain relatively similar in viewing angles, and the observed artifacts are primarily caused by the "invisibility issue", where test views fall outside the input observation sphere, rather than from significant viewpoint deviations. In contrast, our approach tackles large viewpoint shifts without addressing invisibility, emphasizing generalization across substantial angle deviations.

**Regularization Techniques for Unconstrained Reconstruction.** Sparse input views significantly degrade NVS performance, leading to various works exploring geometric priors, spatial regularity constraints, and data-driven priors. *Geometric priors* have been used in SuGaR (Guédon & Lepetit, 2024), 2DGS (Huang et al., 2024a), and GeoGaussian (Li et al., 2024b), which apply handcrafted self-supervision losses to better align Gaussian splats with surface geometry. Similarly, *Spatial regularity* constraints, explored in SplatFields (Mihajlovic et al., 2024) and ZeroRF (Shi et al., 2024), integrate Deep Image Prior (Ulyanov et al., 2018) to regularize 3DGS and NeRF reconstructions, producing more robust results from sparse inputs. However, these methods offer limited improvements as they do not leverage external data. *Data-driven priors* have been adopted in several works. FSGS (Zhu et al., 2024) and DNGaussian (Li et al., 2024a) supervise depth maps using deep stereo models but suffer from scale ambiguity. InstantSplat (Fan et al., 2024a) uses dense point clouds for 3DGS initialization, though it struggles with overfitting. Nerfbusters (Warburg et al., 2023) pretrains a diffusion model for post-processing NeRF, achieving only marginal improvements. Appearance priors (Zhu et al., 2024; Sargent et al., 2024; Wu et al., 2024a; Gao et al., 2024; Kwak et al., 2024) use 2D diffusion models to regularize novel view renderings, but often struggle with multi-view consistency. Additionally, SSDNeRF (Chen et al., 2023) and HypNeRF (Sen et al., 2023) pretrain 3D feature grids on object-centric datasets, yet underperform in multi-category experiments.

**Learning-based 2D-to-3D Models.** Another special case of data-driven models involves training feed-forward models on large-scale multi-view image datasets to predict 3D representations from 2D images. Several methods (Liu et al., 2023b; Höllein et al., 2024; Kong et al., 2024; Liu et al., 2023a), et al. fine-tune pretrained diffusion models to generate multi-view images from one or a few input views. PixelNeRF (Yu et al., 2021), MVSplat (Chen et al., 2024b), and related works (Charatan et al., 2024; Wewer et al., 2024) transform 2D image features into NeRF or Gaussian splats. Although these models can learn generic priors from multi-view datasets, they are typically constrained to only a few input views, limiting their ability to fully leverage larger multi-view inputs.

**3D Point Processing Techniques** are central to our work and widely used across 3D tasks. Unlike 2D image features or 3D grids, point clouds are unordered and unevenly distributed, requiring specialized architectures to handle their irregularity and sparsity. Solutions include sparse convolution (Choy et al., 2019), MLPs (Qi et al., 2017), and transformers (Wang, 2023; Yang et al., 2023; Zhao et al., 2021). The point transformer (Zhao et al., 2021; Wu et al., 2022; 2024b), using attention to model spatial relationships, has proven particularly effective. In 3D reconstruction, CVT-xRF (Zhong et al., 2024) employs a 3D transformer to predict ray point attributes based on local neighborhoods, while LSM (Fan et al., 2024b) utilizes a point transformer to predict 3DGS from two input views. Our work is the first to adapt the point transformer for refining 3D Gaussian splats (3DGS), leveraging its ability to capture spatial relationships in irregular point clouds to enhance novel view synthesis fidelity.

## 3 REVIEW: 3D GAUSSIAN SPLATTING (3DGS)

3D Gaussian Splatting (3DGS) encodes a scene using Gaussian splat primitives $\{\mathcal{G}_k\}_{k=1}^{K}$, which are rendered via volume splatting. Each primitive is defined by its mean position $\mathbf{p}_k \in \mathbb{R}^{3 \times 1}$, opacity $\alpha_k \in [0, 1]$, $S$-dimensional spherical harmonics $\mathbf{a}_k \in \mathbb{R}^S$ for modeling view-dependent color $\mathbf{c}_k \in \mathbb{R}^3$, and covariance matrix $\mathbf{\Sigma}_k \in \mathbb{R}^{3 \times 3}$ parameterized via scale $\mathbf{s}_k \in \mathbb{R}^3$ and rotation quaternion $\mathbf{q}_k \in \mathbb{R}^4$ vectors for enforced positive semi-definiteness.

The splats are rendered by projecting them onto an image plane, forming 2D Gaussian distributions:

$$\mathcal{G}_k^{\text{2D}}(\mathbf{x}') \propto \exp(-(\mathbf{x}' - \mathbf{p}_k')^T (\mathbf{\Sigma}_k^{\text{2D}})^{-1}(\mathbf{x}' - \mathbf{p}_k')/2) \,, \tag{1}$$

where $\mathbf{p}_k' \in \mathbb{R}^2$ and $\mathbf{\Sigma}_k^{\text{2D}} \in \mathbb{R}^{2 \times 2}$ are the projected splat center and covariance matrix.

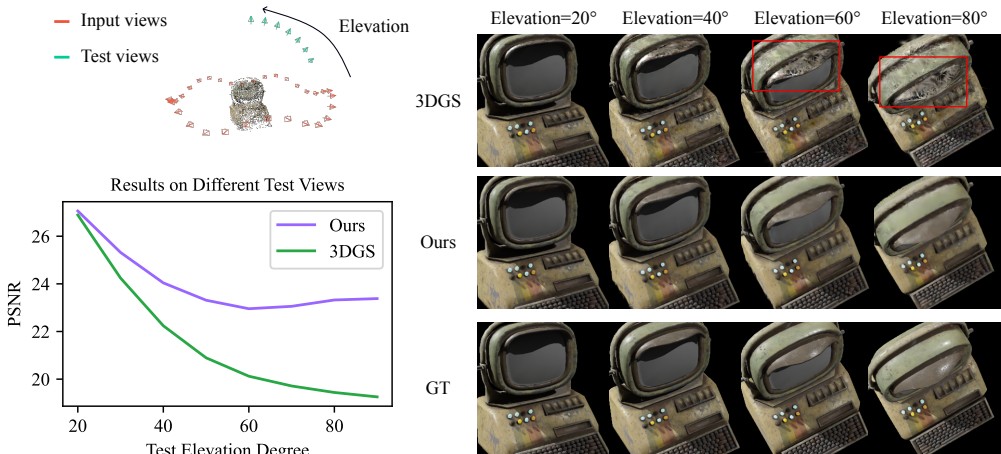

Figure 2: **Limitations of 3DGS in OOD-NVS setup.** We observe that the quality of novel views obtained via 3DGS significantly deteriorates as the test camera deviates from the distribution of input camera views which our solution, *SplatFormer*, effectively overcomes and demonstrates higher fidelity renderings. The displayed metric (left) is performed on the scenes from Objaverse (Deitke et al., 2023); see Sec. 5 for detailed experiment setup.

To compute the pixel color $\mathbf{c}(\mathbf{x}') \in \mathbb{R}^3$ at location $\mathbf{x}' \in \mathbb{R}^2$ splats are blended in sorted depth order:

$$\mathbf{c}(\mathbf{x}') = \sum\nolimits_{k=1}^{K} \mathbf{c}_k \alpha_k \mathcal{G}_k^{2D}(\mathbf{x}') \prod\nolimits_{j=1}^{k-1} (1 - \alpha_j \mathcal{G}_j^{2D}(\mathbf{x}')). \tag{2}$$

*Optimization.* The parameters $\{\mathcal{G}_k\}_{k=1}^{K}$ are optimized using the Adam optimizer (Kingma & Ba, 2015) by minimizing a weighted combination of $\mathcal{L}_1$ and D-SSIM losses:

$$\mathcal{L}_{\text{3DGS}} = (1 - \lambda)\mathcal{L}_1 + \lambda\mathcal{L}_{\text{D-SSIM}}, \tag{3}$$

with $\lambda$ set to 0.2 as per the original 3DGS formulation. To avoid local minima, 3DGS employs periodic heuristic densification and pruning of Gaussian splats.

## 4 ROBUST OUT-OF-DISTRIBUTION NOVEL VIEW SYNTHESIS

**Limitations of 3DGS.** While direct optimization of splat primitives allows 3DGS to closely adapt to input images, it often leads to overfitting, as the flexible primitives conform too precisely to individual pixels. The smooth, continuous nature of Gaussian distributions supports effective interpolation, but only when test views are similar to the training views. To demonstrate this limitation, we conduct a controlled experiment (Fig. 2) simulating a typical scenario where a user captures images while rotating around an object. The challenge arises when rendering from out-of-distribution (OOD) viewpoints, such as elevated camera angles, a critical requirement for AR and VR applications that demand consistent 3D rendering from all perspectives.

**Key Observation.** As shown in Fig. 2, the reconstruction quality degrades significantly as the test camera's elevation increases, highlighting a key limitation of 3DGS in handling OOD views. The challenge is to make the representation robust to such viewpoint changes while preserving the advantages of 3DGS, such as real-time rendering and compatibility with rasterization-based tools. Addressing this limitation by incorporating priors and constraints into the optimization of 3DGS is a complex task. Previous approaches have attempted to address this using geometric constraints (Huang et al., 2024a; Mihajlovic et al., 2024) and data-driven priors (Fan et al., 2024a). However, as demonstrated later (Tab. 1), these methods fall short in achieving robust novel-view synthesis, emphasizing the need for a more effective solution. We believe that solving this issue requires incorporating three key aspects: leveraging generic priors from large-scale datasets, ensuring 3D consistency in renderings, and fully utilizing rich geometric information from all input views.

**Solution: SplatFormer.** We introduce *SplatFormer*, a novel learning-based feed-forward 3D neural module to operate on Gaussian splats, enabling robust novel view synthesis from OOD views. As

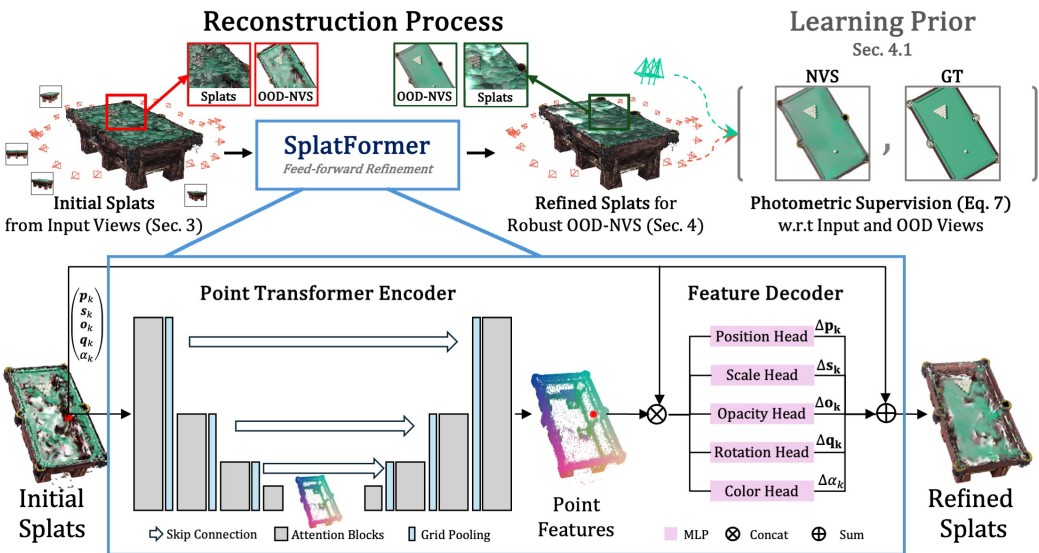

Figure 3: **Method Overview**. We introduce *SplatFormer*, a generalizable 3D point transformer network designed for feed-forward refinement of Gaussian splats, enabling robust out-of-distribution novel-view synthesis (OOD-NVS). The reconstruction process begins by generating an initial set of 3D Gaussians from input images. However, these splats are biased toward the input views and are not robust for OOD-NVS. *SplatFormer* refines these splats through a hierarchical neural network that models residuals to the initial splat attributes. The model is trained on a large collection of 3D shapes using 2D rendering loss, allowing it to: *1)* incorporate spatial regularity among splat primitives via the hierarchical architecture, *2)* leverage generic priors from large-scale datasets, and *3)* ensure 3D consistency through refining 3D primitives directly.

shown in Fig. 2, our method maintains high visual quality even when test views deviate significantly from the input views. SplatFormer, parameterized via learnable parameters $\theta$, overcomes the bias toward input views by capturing spatial relationships and modeling interactions between splats. Inspired by transformer architectures, which excel at learning complex relationships in data (Brown et al., 2020), we adopt this approach for feed-forward refinement of 3D Gaussian splats.

**Reconstruction Process** (Fig. 3) begins with a set of calibrated input images, from which we generate splat primitives $\{\mathcal{G}_k\}_{k=1}^{K}$ using the 3DGS optimization process described in Sec. 3. Since these splats are biased toward the input views, we apply SplatFormer for feed-forward refinement to enable robust out-of-distribution novel-view synthesis. SplatFormer utilizes a hierarchical series of transformer encoder-decoder layers $f_\theta$ based on the Point Transformer V3 (PTv3) architecture (Zhao et al., 2021; Wu et al., 2024b) and is trained on a large collection of 3D shapes using 2D rendering loss. This supervision refines the splat primitives by enforcing spatial regularity through the hierarchical network architecture, leveraging generic priors from large-scale datasets, and ensuring 3D consistency in the refined splats through multi-view consistent rendering supervision. During encoding, SplatFormer assigns each splat an abstract V-dimensional feature vectors $\mathbf{v}_k \in \mathbb{R}^V$:

$$\{\mathbf{v}_k\}_{k=1}^{K} = f_\theta(\{\mathcal{G}_k\}_{k=1}^{K}), \tag{4}$$

which encapsulate key details of the 3D primitives. The feature decoder $g_\theta$ then transforms this latent representation into splat attribute residuals

$$\{\Delta\mathcal{G}_k = (\Delta\mathbf{p}_k, \Delta\mathbf{s}_k, \Delta\alpha_k, \Delta\mathbf{q}_k, \Delta\mathbf{a}_k)\}_{k=1}^{K} = f_\theta(\{\mathcal{G}_k, \mathbf{v}_k\}_{k=1}^{K}), \tag{5}$$

which yields a refined set of splats $\{\mathcal{G}'_k\}_{k=1}^{K}$ that is more robust for OOD novel-view synthesis:

$$\{\mathcal{G}'_k\}_{k=1}^{K} = \{\mathcal{G}_k + \Delta\mathcal{G}_k\}_{k=1}^{K}. \tag{6}$$

**Point Transformer Encoder** $f_\theta$. Our 3DGS splat encoder is based on the PTv3 framework (Wu et al., 2024b). The input set of points is first passed through an embedding layer to obtain corresponding input features, followed by 5 stages of attention blocks and downsampling grid pooling

Table 1: **OOD-NVS.** Comparisons on the ShapeNet-OOD and Objaverse-OOD evaluation sets. The metric is evaluated on OOD test views with elevation $\phi_{\text{ood}} \geq 70°$; colors indicate the 1st , 2nd , and 3rd best-performing model

| | Methods | ShapeNet-OOD | | | Objverse-OOD | | |
|---|---|---|---|---|---|---|---|
| | | PSNR | SSIM | LPIPS | PSNR | SSIM | LPIPS |
| Standard | MipNeRF360 (Barron et al., 2022) | 20.06 | 0.819 | 0.265 | 19.64 | 0.722 | 0.280 |
| | InstantNGP (Müller et al., 2022) | 17.09 | 0.684 | 0.339 | 19.47 | 0.694 | 0.310 |
| | 3DGS (Kerbl et al., 2023) | 20.21 | 0.763 | 0.242 | 19.24 | 0.673 | 0.285 |
| Regularized | 2DGS (Huang et al., 2024a) | 23.52 | 0.863 | 0.188 | 20.56 | 0.739 | 0.248 |
| | SplatFields (Mihajlovic et al., 2024) | 23.15 | 0.850 | 0.185 | 18.85 | 0.688 | 0.308 |
| External Prior | InstantSplat (Fan et al., 2024a) | 18.49 | 0.735 | 0.304 | 15.61 | 0.523 | 0.437 |
| | FSGS (Zhu et al., 2024) | 17.32 | 0.714 | 0.298 | 18.82 | 0.655 | 0.323 |
| | SSDNeRF (Chen et al., 2023) | 15.36 | 0.650 | 0.393 | 16.90 | 0.552 | 0.434 |
| | Nerfbusters (Warburg et al., 2023) | 11.42 | 0.640 | 0.321 | 16.87 | 0.689 | 0.287 |
| Feed Forward | SyncDreamer (Liu et al., 2023b) | 17.26 | 0.738 | 0.316 | 12.47 | 0.542 | 0.384 |
| | EscherNet (Kong et al., 2024) | 19.63 | 0.786 | 0.236 | 16.09 | 0.609 | 0.263 |
| | LaRa (Chen et al., 2024a) | 20.94 | 0.839 | 0.222 | 19.04 | 0.682 | 0.324 |
| | SplatFormer | 27.98 | 0.920 | 0.136 | 23.06 | 0.821 | 0.170 |

layers (Wu et al., 2022). Then another 4 stages of attention blocks and upsampling grid pooling layers are used to restore the resolution. To capture high-frequency details and improve gradient flow, skip connection MLP modules are used to map intermediate downsampling outputs to residuals, which are then added to the upsampling layers at corresponding resolutions. Each stage comprises attention blocks with layer normalization, multi-head attention, and MLPs. This hierarchical architecture models contextual relationships among neighboring primitives. To implement attention efficiently based on spatial proximity, we adopt PTv3's serialization and grid pooling strategy.

**Feature Decoder** $g_\theta$. The extracted features are further concatenated with the original splat attributes to enhance convergence by combining the transformer's context-aware features with the initial attributes. Each point's features are then passed into shared feature decoding heads, which consist of five sequential MLP modules to predict residuals to the initial splat attributes. To further improve training stability, we zero-initialize the final MLP layers' weights and biases leading to zero initial residual features, ensuring that the initial output closely matches the input 3DGS.

## 4.1 LEARNING DATA-DRIVEN PRIOR

**Dataset.** To enable SplatFormer to refine imperfect Gaussian splats using a data-driven prior, we curated a large dataset containing pairs of Gaussian primitives and corresponding multi-view images. We utilized 33k and 48k scenes from the ShapeNet (Chang et al., 2015) and Objaverse-1.0 (Deitke et al., 2023) datasets respectively. These assets were rendered from low-elevation input views and high-elevation out-of-distribution (OOD) views. The initial splats were generated from the low-elevation views (following Sec. 3). The data collection process, which required approximately 3000 GPU hours, was efficiently executed using budget GPUs like the RTX-2080Ti. We released the data and corresponding rendering code to facilitate future research.

**Training Objective.** After generating the initial 3DGS by minimizing the photometric loss (Eq. 3) using low-elevation input views, the SplatFormer module performs feed-forward refinement. The refined splats are then rendered following Eq. 2 for both input and OOD views, using a combination of photometric and perceptual LPIPS (Zhang et al., 2018) loss terms:

$$\mathcal{L}_{\text{SplatFormer}} = \mathcal{L}_1 + \mathcal{L}_{\text{LPIPS}} . \tag{7}$$

This loss is optimized using the Adam optimizer (Kingma & Ba, 2015) across multi-view images, incorporating both low-elevation and high-elevation OOD views. This balanced approach ensures that the model generalizes to unseen angles while preserving high fidelity for in-distribution views.

The dataset and training approach we introduce allow SplatFormer to learn rich data-driven priors from a diverse range of 3D objects and view configurations. These learned priors enable the model to correct 3DGS's bias towards input views, leading to more accurate and consistent reconstructions in OOD scenarios.

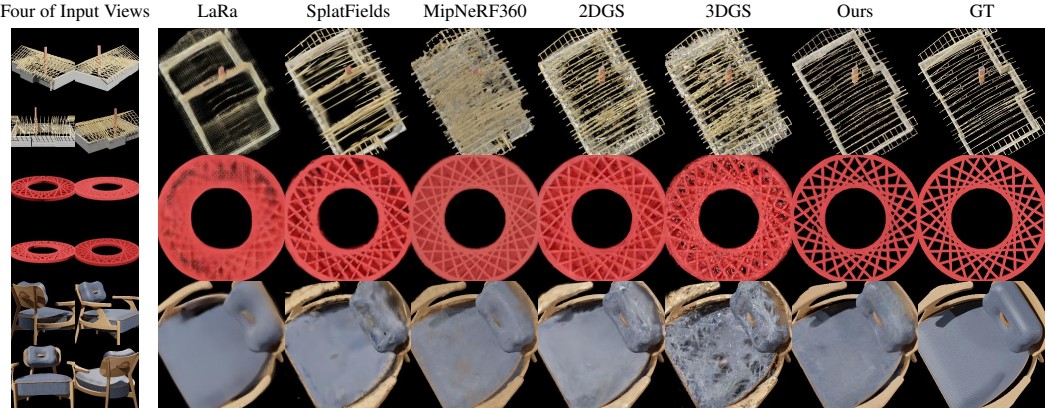

Figure 4: **Novel View Synthesis under Out-of-Distribution Camera Angles.** The first column shows *4 out of 32* input views. Here, we compare our method with LaRa (Chen et al., 2024a), Splat-Fields (Mihajlovic et al., 2024), MipNeRF360 (Barron et al., 2022), 2DGS (Huang et al., 2024a), and 3DGS (Kerbl et al., 2023). Results on Objaverse-OOD evaluation scenes; a comprehensive comparison with all the baselines is provided in the appendix (Fig. H.4).

## 5 EXPERIMENTS

We begin by outlining our experimental setup, followed by a description of the evaluation protocol and the baseline methods used for comparison. Next, we present the results on OOD-NVS, cross-dataset generalization, and ablation studies. Finally, we discuss the limitations of our approach and potential directions for future research.

**OOD-NVS.** As visualized in Fig 1, for a centered object, we simulate an input camera capturing 360-degree azimuths at low elevations. The camera takes $N_{in}$ photos from evenly spaced azimuths, with its elevation following a sinusoidal pattern defined by frequency $f$ and maximal elevation $\phi_{max}$. This mimics a user recording the target with physical constraints preventing stable top-down captures. For OOD test views, we set their elevations $\phi_{ood} \gg \phi_{max}$, simulating a top-down perspective.

**Evaluation Sets.** We use Blender to render 20 objects from ShapeNet (Chang et al., 2015) and Objaverse-v1 (Deitke et al., 2023) each. We select common objects and scenes with meaningful top-down views, such as city streets and buildings, avoiding those with large cavities that are invisible from low-elevation views. Due to varying object heights and shapes, we render two input camera trajectories with $\phi_{max} = 10°$ and $20°$, creating two input-target splits to represent different levels of view deviation. So each scene has two experiments with different input views and the same target views. We average evaluation metrics across the two sets of experiments. Each input trajectory consists of $N_{in} = 32$ views. The OOD test set includes $N_{out} = 9$ views, uniformly distributed from the top sphere with $\phi_{ood} \geq 70°$. All renderings are at a resolution of $256 \times 256$.

**Baselines.** We evaluate SplatFormer using the OOD-NVS protocol, comparing it with several publicly available state-of-the-art methods from each method category. First, we include per-scene approaches designed primarily for in-distribution NVS, including InstantNGP (Müller et al., 2022), 3DGS (Kerbl et al., 2023), and MipNeRF360 (Barron et al., 2022). Next, we examine regularized 3DGS variants without external priors, including 2DGS (Huang et al., 2024a) and SplatFields (Mihajlovic et al., 2024). We further consider methods which regularize per-scene optimization using external priors, including InstantSplat (Fan et al., 2024a), FSGS (Zhu et al., 2024), SSDNeRF (Chen et al., 2023), and Nerfbusters (Warburg et al., 2023). Finally, we examine feed-forward models, which directly produce 2D Gaussians (LaRa (Chen et al., 2024a)) or multi-view images (Sync-Dreamer (Liu et al., 2023b) and EscherNet (Kong et al., 2024)) from input images.

For a fair comparison, we retrained SSDNeRF, LaRa, SyncDreamer, and EscherNet using the same training sets as SplatFormer. For SyncDreamer, which only supports single-view input, we selected the highest-elevation view. For EscherNet, we use three randomly sampled input views during training and all input views during inference. For LaRa, which is limited to four input views due to memory constraints, we chose four large-baseline views to maximize scene coverage. More details are provided in appendix (Sec. G).

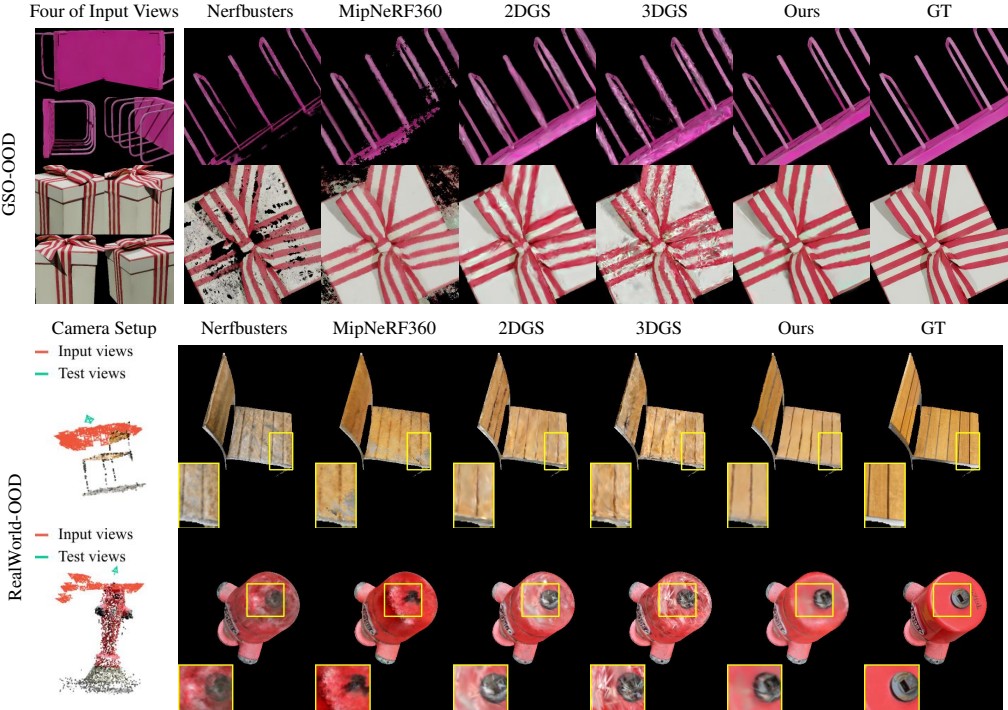

Figure 5: **Cross-dataset Generalization.** SplatFormer trained on Objaverse successfully mitigates artifacts in OOD views in the GSO (Downs et al., 2022) dataset and our real-world object-centric captures. Additional results are presented in the appendix (Fig. H.5 and Fig. H.6).

**Results on OOD-NVS.** Qualitative results (Fig. 4) show that LaRa produces blurry outputs, while MipNeRF360 suffers from floater artifacts and SplatFields smooths out fine details. Both 2DGS and 3DGS exhibit spiky artifacts. In contrast, SplatFormer significantly reduces the artifacts present in 3DGS, completes surface reconstruction, and even restores certain geometric properties, such as interlaced structures. While our method still faces challenges with high-frequency texture details, it outperforms previous approaches in terms of fidelity and consistency in out-of-distribution views, which is also supported by the clear quantitative improvements demonstrated in our results (Tab. 1). Additional visual results are provided in the appendix (Sec. F) and the supplementary video.

**Generalization Across a Range of View Deviations.** Our method does not overfit to the extreme top views present in the SplatFormer training set but generalizes across a range of views, transitioning from input to extreme target views. To demonstrate this, we evaluate NVS with elevations $\phi \in [10°, 90°]$ in Objaverse-OOD scenes and compare SplatFormer to 3DGS (Fig. 2). While 3DGS performance degrades significantly as the viewing angle deviates from the input views, our method provides more robust synthesis for target views in the range $\phi \in [25°, 90°]$.

**Cross-dataset, Real World Generalization.** Following the OOD-NVS protocol, we rendered 20 objects from Google Scanned Objects (GSO) (Downs et al., 2022) and captured 4 real-world scenes. Low-elevation views were used to optimize the initial set of Gaussians, while the OOD views were reserved for validation. For real-world captures with unbounded backgrounds, we segmented the foreground objects. Additional details are provided in the

Table 2: **Cross-dataset Generalization**

| Methods | GSO-OOD | | | RealWorld-OOD | | |
|---|---|---|---|---|---|---|
| | PSNR | SSIM | LPIPS | PSNR | SSIM | LPIPS |
| Nerfbusters | 15.95 | 0.678 | 0.300 | 23.93 | 0.893 | 0.114 |
| 2DGS | 23.29 | 0.816 | 0.204 | 23.64 | 0.891 | 0.104 |
| MipNeRF360 | 22.90 | 0.824 | 0.192 | 21.99 | 0.878 | 0.127 |
| 3DGS | 21.78 | 0.746 | 0.250 | 23.83 | 0.877 | 0.109 |
| SplatFormer | **25.01** | **0.863** | **0.148** | **24.33** | **0.902** | **0.100** |

appendix (Sec. A.1). SplatFormer, trained on synthetic data, shows generalization to 3D-scanned real-world objects from the GSO dataset, as well as to real-world mobile phone captures (Tab. 2, Fig. 5). This suggests that our method does not learn object-specific prior as SSDNeRF (Chen et al., 2023) and HypNeRF (Sen et al., 2023), and the 3DGS refinement prior can be transferred across object categories.

On the GSO-OOD evaluation set, SplatFormer achieves substantial improvements in both metrics and visual quality. Even on the real-world dataset, despite being trained exclusively on synthetic data, SplatFormer reduces artifacts. These improvements are reflected in the SSIM and LPIPS metrics, though we observed rather minimal improvements in PSNR, which we attribute to the pixel-wise PSNR's sensitivity to imperfect calibration and our method's limitation in modeling specular effects. Our method also outperforms MipNeRF360 and 2DGS, the best-performing baselines in Objaverse-OOD (Tab. 1). Additionally, we evaluate Nerfbusters (Warburg et al., 2023), which addresses robust novel view synthesis from novel camera trajectories—a relevant challenge to OOD-NVS. However, we find that Nerfbusters tends to mistakenly remove key scene content as floaters, leading to incomplete geometry.

**3D *vs* 2D Denoising.** An alternative strategy for refining OOD-NVS renderings is to use 2D image restoration methods. To explore this, we use Diff-BIR (Lin et al., 2021), a state-of-the-art image restoration method, to denoise 3DGS renderings. DiffBIR consists of two cascaded models: a first-stage image-to-image regressor to remove artifacts and a second-stage diffusion-based generator (Rombach et al., 2021) to in-paint missing details. We trained both stages us-

Table 3: **SplatFormer *vs* 2D Denoising**

| Method | ShapeNet-OOD | | |
| --- | --- | --- | --- |
| | PSNR | SSIM | LPIPS |
| 3DGS | 20.21 | 0.763 | 0.242 |
| DiffBIR-stage1 (Lin et al., 2021) | 24.81 | 0.892 | 0.163 |
| DiffBIR-stage2 (Lin et al., 2021) | 24.24 | 0.858 | 0.174 |
| Retrained 3DGS with stage1 | 25.16 | 0.894 | 0.164 |
| Retrained 3DGS with stage2 | 24.83 | 0.870 | 0.174 |
| SplatFormer | **28.09** | **0.920** | **0.135** |

ing pairs of OOD 3DGS renderings and ground-truth images from our ShapeNet-OOD training set. To address multi-view inconsistencies in the denoised images, we also used the generated images to retrain the 3DGS. This experiment is similar to Sp2360 (Paul et al., 2024), which uses cascaded 2D diffusion priors to regularize 3DGS from sparse-view inputs. As shown in Tab. 3, while 2D denoising methods improve the original 3DGS, they significantly underperform compared to SplatFormer and fail to recover geometric details. See the appendix (Fig. C.1) for visual comparisons.

**Ablation: Backbone and Supervision.** We compare our PTv3 (Wu et al., 2024b) transformer-based architecture with widely used Minkowski (Choy et al., 2019) engine. Additionally, to validate the effectiveness of the residual prediction strategy outlined in Sec. 4, we train a variant that directly predicts the full 3DGS attributes

Table 4: **Ablations**

| Backbone | Prediction | Objaverse-OOD | | |
| --- | --- | --- | --- | --- |
| | | PSNR | SSIM | LPIPS |
| PTv3 (Wu et al., 2024b) | Direct | 21.36 | 0.772 | 0.211 |
| Mink (Choy et al., 2019) | Residual | 22.67 | 0.807 | 0.181 |
| PTv3 (Wu et al., 2024b) | Residual | **23.06** | **0.821** | **0.170** |

(direct component). The results in Tab. 4 show that the point transformer architecture and residual-based learning improve performance compared to the alternatives.

**Limitations and Future Work.** Our method has several limitations that provide directions for future work. First, despite outperforming all the considered baselines, it still struggles to reconstruct fine-grained details and complex texture. Second, the generalization to real-world captures could be improved by scaling up training examples and by enhancing the realism of synthetic lighting. Third, applying our method to refining 2DGS may further improve the OOD-NVS results. Finally, it would be valuable to train our method to remove OOD-NVS artifacts in unbounded scenes and with a wider range of OOD camera setups. In the appendix H, we present a experimental result in Fig. H.1 and Tab. H.1 using the MVImgNet dataset (Yu et al., 2023), and outline both the potential and challenges. Please refer to it for an extended discussion.

## 6 CONCLUSION

Photorealistic rendering of 3D assets under diverse viewing conditions is critical for AR and VR applications. In this work, we introduced a new out-of-distribution (OOD) novel view synthesis test scenario and demonstrated that most neural rendering methods, including those using regularization techniques and data-driven priors, suffer substantial quality degradation when test viewing angles deviate significantly from the training set, highlighting the need for more robust rendering techniques. As an initial step towards addressing the problem, we proposed SplatFormer, a novel point transformer model designed to overcome the limitations of 3D Gaussian Splatting in handling OOD views. By refining 3DGS representations in a single forward pass, SplatFormer significantly improves rendering quality in these scenarios and achieves state-of-the-art performance, outperforming prior methods designed for both sparse and dense view inputs. The success of our model further underscores the potential of integrating transformers into photorealistic rendering workflows.

ACKNOWLEDGEMENTS

This study was conducted within the national "Proficiency"[1] research project funded by the Swiss Innovation Agency Innosuisse in 2021 as one of 15 flagship initiatives. This work was also supported as part of the Swiss AI Initiative by a grant from the Swiss National Supercomputing Centre (CSCS) under project ID a03 on Alps. Marko Mihajlovic is in part supported by the Hasler Stiftung Grant (2024-09-12-159).

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

APPENDIX: SPLATFORMER: POINT TRANSFORMER FOR ROBUST 3D GAUSSIAN SPLATTING

We provide details on the evaluation datasets (Sec. A) and implementations of our method (Sec. B). Then, we show more experimental results, including ablation studies (Sec. C), evaluation on geometry (Sec. D), evaluation on a diverse range of test views (Sec. D) and comprehensive visual comparisons (Sec. F). Additionally, we describe baseline implementations (Sec. G). Finally, we discuss the limitations of SplatFormer in Sec. H.

## A  EVALUATION DATASETS

**Synthetic Datasets.**  We use Blender to render objects from ShapeNet (Chang et al., 2015), Objaverse-v1 (Deitke et al., 2023), and GSO (Google Scanned Objects) (Downs et al., 2022). The camera setups for the three evaluation sets are consistent: $N_{\text{in}} = 32$ input views cover the $360°$ azimuth and elevation angle $\phi$ varies according to a sinusoidal function ranging between $(0, \phi_{\text{max}})$. For each object in ShapeNet, we rotate the object's shortest side with the z-axis and render a single set of input views with $\phi_{\text{max}} = 10°$. For each object in Objaverse-v1 and GSO, we render two sets of input views with $\phi_{\text{max}} = 10°, 20°$. The out-of-distribution (OOD) test set consists of $N_{\text{out}} = 9$ views, with $\phi_{\text{ood}} = (70°, 80°, 90°)$ and uniformly strided azimuths. The rendered resolution is $256 \times 256$ pixels. The resulting ShapeNet-OOD, Objaverse-OOD, GSO-OOD datasets include a total of 20, 40, and 40 input-test experiments, respectively. We enable specular effects to achieve more realistic rendering results when using objects from Objaverse-v1 and GSO, and disable specular reflections for ShapeNet to study a more basic illumination setup.

**Real-world OOD iPhone Dataset.** We have captured 4 scenes featuring an object of interest using an iPhone, with the images and camera setups shown in Fig. H.6. Each scene contains around 30 input views and 4 OOD test views. During evaluation, we first generate foreground masks of the objects of interest for the OOD test view using SAM2 (Ravi et al., 2024), and then only evaluate the pixels within the mask. To refine the 3DGS representation via SplatFormer, we crop out the part of the 3DGS point cloud that corresponds to the foreground region using selection tools in MeshLab. This may also be easily done by automatic 3D detection methods like Segment3D (Huang et al., 2024b). The cropped splats are then refined via SplatFormer and rendered using the standard 3DGS-based rendering pipeline. We resize images to the resolution of $300 \times 400$ for both 3DGS training and evaluation.

We present examples from the four datasets, as well as the degraded 3DGS OOD renderings, in Fig. A.1. It is worth noting that the dense input capture covers a substantial portion of the objects, eliminating the need for novel view synthesis (NVS) methods to hallucinate unobserved parts during target view generation.

## B  IMPLEMENTATION DETAILS

**Network Arhitecture.** The point transformer encoder begins with an MLP embedding layer, followed by five down-pooling and four up-pooling stages, ultimately producing features with a dimensionality of $V = 96$. The down-pooling stages contain $(2, 2, 2, 6, 2)$ attention blocks and have hidden dimensions of $(64, 96, 128, 256, 512)$. Each down-pooling stage, except the first, is followed by a down-sampling grid-pooling layer. The up-pooling stages consist of $(2, 2, 2, 2)$ attention blocks, with hidden dimensions of $(256, 128, 96, 96)$. Each up-pooling stage, except the last, is preceded by an up-sampling grid-pooling layer. A grid resolution of 384 is used to voxelize the point cloud, and the strides for the grid-pooling layers are set to $(1, 2, 2, 2)$. For the architecture details of attention blocks and grid pooling please refer to Wu et al. (2024b).

The feature decoder is composed of five separate MLP branches, which are responsible for predicting the residuals for the means, opacity, quaternion, scales, and spherical harmonics coefficients. Each MLP branch consists of four linear layers, with hidden dimensions of 512 and ReLU activations for all but the last layer. Tanh activation is applied to normalize the residual means to the range $[0, 1]$. Additionally, the positions of the input 3D Gaussians are normalized to $[0, 1]^3$. The total number of parameters is approximately 50 million.

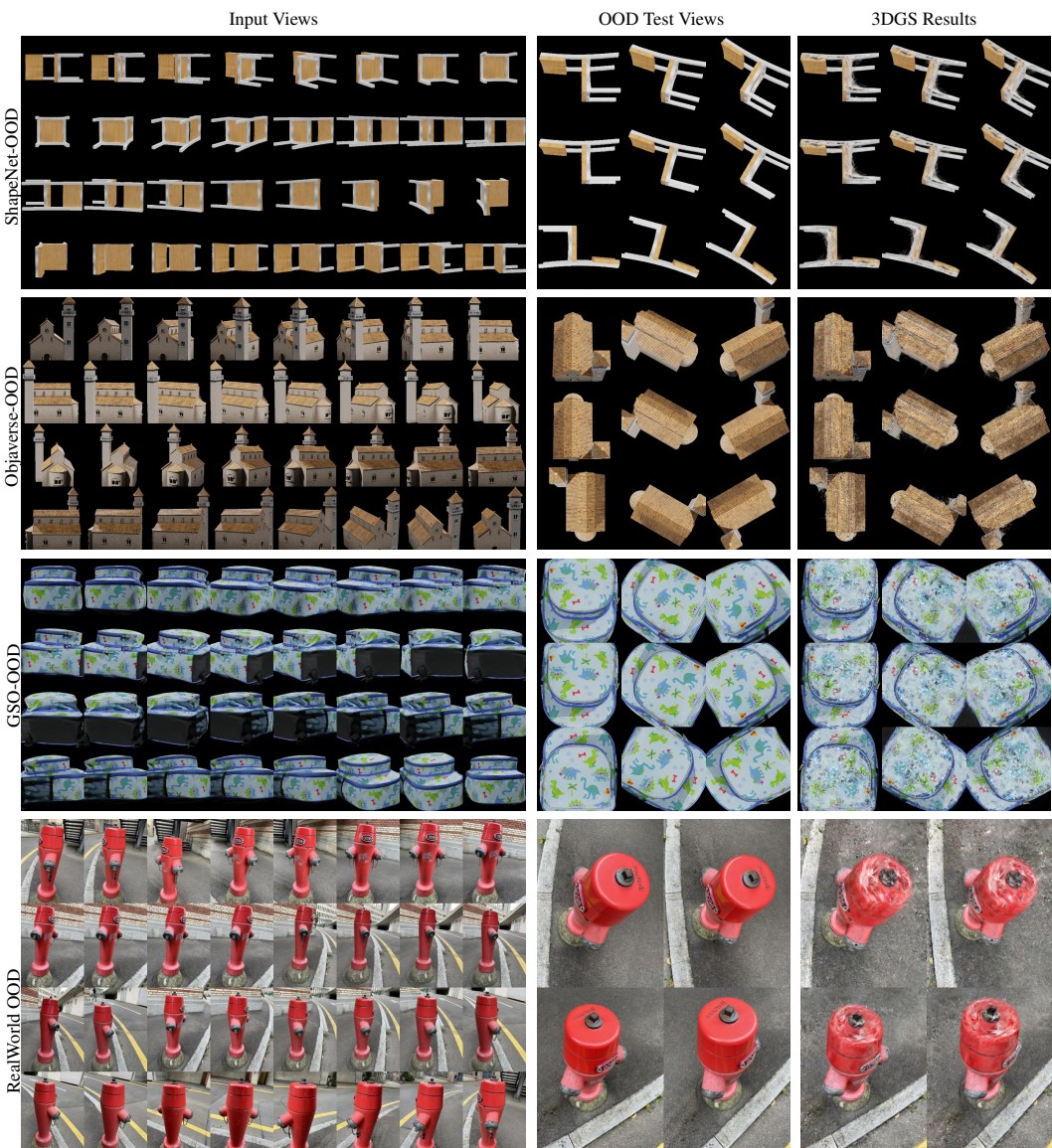

Figure A.1: **Examples from our OOD-NVS evaluation sets and the artifacts in 3DGS.**

**Training Dataset Curation.** The ShapeNet training set contains 33k objects, all available for non-commercial research and educational purposes. The Objaverse training set includes 48k objects from Objaverse-1.0 (Deitke et al., 2023), all licensed under Creative Commons for distribution. We use Blender to render each object with 32 low-elevation views and 5 top-down views. Diffuse lighting and materials are applied in ShapeNet scenes, while specular effects and shadows are enabled in Objaverse scenes. For the rendered 2D low-elevation views, we use gsplat (Ye et al., 2024) to optimize the initial 3D Gaussian splats (3DGS) for each scene. The spherical harmonics degree is set to 0 for ShapeNet and 1 for Objaverse. To reduce computational costs, we terminate the optimization early at 10k steps, where evaluation performance levels off. We process the scenes using 48 RTX-2080Ti GPUs, with rendering and 3DGS optimization taking approximately 3 minutes per scene. It takes 2 days to generate each training dataset.

**Training.** For each scene, we render 4 target images at each iteration, with 70% OOD views and 30% input views, for photometric supervision. For the training of our full model, we use 8 RTX-4090s with one scene per GPU, set gradient accumulation steps as 4, and train for 150k iterations, which takes around 2 days. We use Adam optimizer with a constant learning rate of 3e-5. During

training, we cap the number of input Gaussians to SplatFormer at 100k. If the 3DGS exceeds this threshold, we randomly subsample the Gaussians.

**Inference.** To obtain the refined 3DGS for each test scene, we first train a 3DGS with 32 input views using gsplat (Ye et al., 2024) for 10k steps. Then, we feed the 3DGS into SplatFormer to obtain the refined output. Regarding SplatFormer's inference efficiency, most input splats in our object-centric test sets contain 70k—100k gaussians, requiring only 900MB of GPU memory for one feed-forward inference pass and achieving an inference time of 108 ms. To evaluate the upper bound of SplatFormer's inference capability, we increase the number of Gaussians by sampling additional Gaussians with Gaussian noise. We find that an RTX 4090 GPU can accommodate up to 4 million Gaussians. However, it is important to note that the GPU memory consumption of the point transformer is not solely determined by the number of input points. Instead, it is also significantly influenced by the spatial distribution of the points. A 3DGS with a spatially uniform distribution and high entropy tends to consume more GPU memory than one with a more concentrated distribution. Since object-centric scenes often possess concentrated spatial distribution, our current SplatFormer can be quite efficient for large-scale 3DGS during inference. The primary computational bottleneck still lies in the training stage. Further improving the efficiency of point transformer for large-scale unbounded scenes remains an important direction for future work.

## C  ADDITIONAL ABLATION STUDIES

**SplatFormer *vs* 2D Denoising.** In addition to the metrics in Tab. 3, Fig. C.1 presents a visual comparison between the 2D denoising method DiffBIR (Lin et al., 2021) and SplatFormer on ShapeNet-OOD test views. Though the DiffBIR-stage1 model removes certain artifacts, the improvements are inconsistent across views. Retraining a 3DGS model using the generated images fails to fully address these limitations. Additionally, the stage-1 model struggles to infer correct geometry from the noisy 2D images, causing errors that propagate to stage-2, which may introduce unfaithful hallucinations. In contrast, our method processes both input and output in 3D, resulting in more accurate and consistent artifact removal.

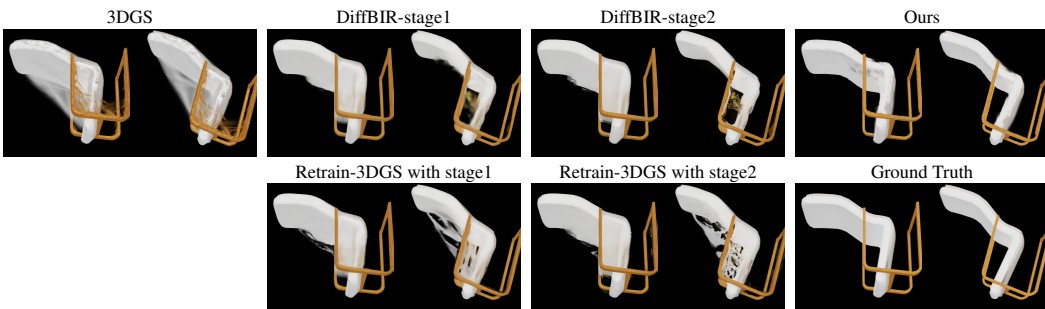

Figure C.1: We adopt DiffBIR (Lin et al., 2021) to denoise artifacts in 2D space. Additionally, we retrain 3DGS using the denoised images to improve multi-view consistency. However, 2D denoising alone is insufficient for fully recovering geometry, as it relies solely on 2D inputs.

**3D *vs* 2D supervision.** We use photometric supervision (Eq. 7) to train our model. An alternative training approach involves supervising the output of SplatFormer with direct 3D labels, *e.g.* such as an optimal 3DGS trained using full-degree views observation. As shown in Fig. C.2, we find that the 3D direct supervision does not consistently enhance the results and presents several limitations. First, it is time consuming to prepare full-degree renderings, making it impractical to scale up the training dataset. Second, it is impossible to train a neural network to fit the 3D signals with 100% accuracy due to the spectral bias (Rahaman et al., 2019), and small errors in 3D prediction can still lead to significant 2D artifacts.

To demonstrate this, we conduct a toy experiment where we overfit the 20 scenes of the ShapeNet-OOD evaluation set. For each scene, besides the flawed 3DGS trained on low-elevation input views,

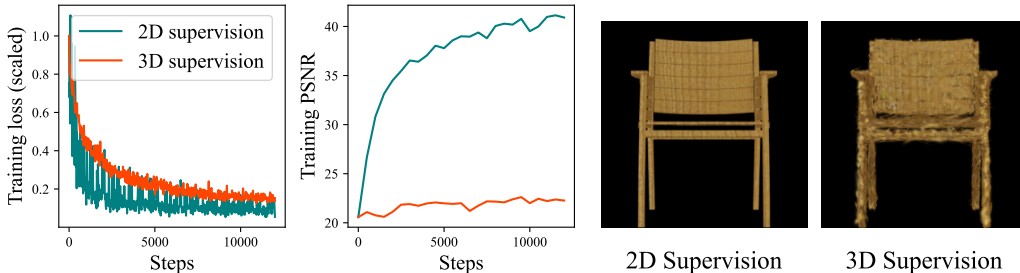

Figure C.2: We overfit two SplatFormers on 20 scenes with 2D partial or 3D direct supervision. We show the training curves and the OOD-view rendering of a training example. Minimizing 3D loss does not improve PSNR of the 2D renderings. Without fitting the 3D label with 100% accuracy, the model with 3D supervision cannot remove artifacts in 2D renderings.

we render 24 views from the upper hemi-sphere. We combine the 24 upper views and the 32 lower views as input views to optimize the flawed 3DGS that is initially trained with the lower views. We disable densification and prunning during the 3DGS optimization so as to ensure the one-to-one correspondence between the input 3DGS and the optimal one. The yielded 3DGS can serve as a pseudo 3D label for SplatFormer training. Then we train two SplatFormers using the 20 scenes. One is trained with photometric loss and the other is trained with the L1 norm error between the output 3D attributes and the pseudo 3D labels. Fig. C.2 shows that while both training objectives can be minimized, only 2D supervision can lead to the improvement in the rendering quality. Therefore, we employ 2D supervision to train SplatFormer, which enhances rendering quality and improves efficiency.

## D    GEOMETRY RESULTS

Building upon 3DGS, SplatFormer focuses primarily on enhancing novel view synthesis rather than surface extraction. However, we demonstrate that our method can still refine the geometry of input 3D Gaussians. Tab. D.1 compares the mean absolute errors (MAE) of rendered depths and normals under out-of-distribution (OOD) views between 3DGS and our approach. Specifically, depth maps for both methods are obtained as the weighted average depth of Gaussian primitives, follow-

Table D.1: **Geometry Evaluation on Objaverse-OOD.**

|      | Depth-MAE↓ | Normal-MAE↓ |
|------|------------|-------------|
| 3DGS | 6.70e-4    | 0.239       |
| Ours | **4.05e-4**| **0.214**   |

ing the standard approach implemented in the gsplat toolbox (Ye et al., 2024). Normal maps are then computed using finite differences on the estimated surface derived from the depth maps, as in 2DGS (Huang et al., 2024a). Fig. H.3 visualizes the depth and normal estimations, highlighting both quantitative and qualitative improvements achieved by our method over 3DGS.

## E    EVALUATION ACROSS DIVERSE TEST VIEWS

Though Tab. 1 and Tab. 2 only evaluate views with elevation $\phi_{\text{ood}} = (70°, 80°, 90°)$ and camera-to-origin distance $R = 1$, SplatFormer can also enhance a wide range of test views with various elevations and even extreme close-up views. Fig. 2 shows that SplatFormer consistently outperforms 3DGS at elevation angles between $20°$ and $90°$. To further demonstrate this, we compare the PSNRs for views with different elevations $\phi \in [20°, 90°]$ and camera radii ($R \in [0.2, 1.0]$) between 3DGS and SplatFormer on the GSO-OOD dataset. As shown in Tab. E.1, SplatFormer significantly outperforms 3DGS across various viewing angles and even in extreme close-up views ($R = 0.2$).

## F    COMPARISONS WITH BASELINES

**Evaluation Details.** As mentioned in Sec. A, we create two experimental setups for each scene in Objaverse and GSO, with the same test views yet different input views with maximum elevations

Table E.1: **Results on various test views.** We evaluate the PSNRs on novel views with various elevation angles $\phi$ and camera-to-origin distance $R$ in GSO-OOD test sets. Our method, SplatFormer, outperforms 3DGS (Kerbl et al., 2023) across various viewing angles and even in zoomed-in views.

| | | $\phi = 20°$ | $\phi = 30°$ | $\phi = 40°$ | $\phi = 50°$ | $\phi = 60°$ | $\phi = 70°$ | $\phi = 80°$ | $\phi = 90°$ |
|---|---|---|---|---|---|---|---|---|---|
| $R = 0.2$ | 3DGS | 22.29 | 21.13 | 20.73 | 19.17 | 18.60 | 18.00 | 17.46 | 16.84 |
| | Ours | **22.99** | **22.19** | **21.85** | **21.50** | **21.44** | **21.18** | **20.89** | **20.18** |
| $R = 0.4$ | 3DGS | 23.28 | 22.43 | 21.12 | 19.87 | 18.99 | 18.49 | 18.13 | 17.98 |
| | Ours | **23.74** | **23.14** | **22.44** | **21.76** | **21.38** | **21.23** | **21.00** | **21.04** |
| $R = 0.6$ | 3DGS | 25.13 | 23.52 | 22.14 | 21.12 | 20.33 | 19.76 | 19.40 | 19.39 |
| | Ours | **25.33** | **24.04** | **23.19** | **22.71** | **22.43** | **22.24** | **22.14** | **22.22** |
| $R = 0.8$ | 3DGS | **27.85** | 25.86 | 24.07 | 22.74 | 21.89 | 21.28 | 20.89 | 20.85 |
| | Ours | 27.28 | **25.87** | **24.85** | **24.21** | **23.89** | **23.68** | **23.57** | **23.61** |
| $R = 1.0$ | 3DGS | **29.62** | 26.65 | 24.58 | 23.30 | 22.50 | 21.97 | 21.70 | 21.66 |
| | Ours | 28.97 | **27.62** | **26.54** | **25.77** | **25.34** | **25.08** | **24.93** | **25.03** |

Table F.1: **Detailed Results on Objaverse-OOD.** We report the separate OOD evaluation results of the two sets of experiments with input views' maximum elevations $\phi_{\max} = (10°, 20°)$. The average results are used in Tab. 1. Colors indicate the 1st , 2nd , and 3rd best-performing models.

| Methods | $\phi_{max} = 10°$ | | | $\phi_{max} = 20°$ | | | Average | | |
|---|---|---|---|---|---|---|---|---|---|
| | PSNR | SSIM | LPIPS | PSNR | SSIM | LPIPS | PSNR | SSIM | LPIPS |
| MipNeRF360 (Barron et al., 2022) | 18.94 | 0.695 | 0.301 | 20.33 | 0.749 | 0.259 | 19.64 | 0.722 | 0.280 |
| InstantNGP (Müller et al., 2022) | 18.60 | 0.662 | 0.334 | 20.34 | 0.726 | 0.285 | 19.47 | 0.694 | 0.310 |
| 3DGS (Kerbl et al., 2023) | 18.30 | 0.642 | 0.305 | 20.18 | 0.703 | 0.265 | 19.24 | 0.673 | 0.285 |
| 2DGS (Huang et al., 2024a) | 19.75 | 0.712 | 0.269 | 21.38 | 0.766 | 0.226 | 20.56 | 0.739 | 0.248 |
| SplatFields (Mihajlovic et al., 2024) | 18.23 | 0.669 | 0.326 | 19.46 | 0.707 | 0.291 | 18.85 | 0.688 | 0.309 |
| InstantSplat (Fan et al., 2024a) | 13.96 | 0.462 | 0.480 | 17.19 | 0.580 | 0.396 | 15.61 | 0.523 | 0.437 |
| FSGS (Zhu et al., 2024) | 17.95 | 0.625 | 0.349 | 19.69 | 0.685 | 0.297 | 18.82 | 0.655 | 0.323 |
| SSDNeRF (Chen et al., 2023) | 16.22 | 0.523 | 0.454 | 17.58 | 0.580 | 0.414 | 16.90 | 0.552 | 0.434 |
| Nerfbusters (Warburg et al., 2023) | 15.25 | 0.643 | 0.318 | 18.49 | 0.735 | 0.255 | 16.87 | 0.689 | 0.287 |
| SyncDreamer (Liu et al., 2023b) | 12.47 | 0.542 | 0.385 | 12.47 | 0.542 | 0.382 | 12.47 | 0.542 | 0.384 |
| EscherNet (Kong et al., 2024) | 15.99 | 0.606 | 0.267 | 16.20 | 0.611 | 0.258 | 16.09 | 0.608 | 0.263 |
| LaRa (Chen et al., 2024a) | 18.56 | 0.672 | 0.336 | 19.53 | 0.697 | 0.313 | 19.04 | 0.684 | 0.324 |
| SplatFormer | 22.15 | 0.795 | 0.190 | 23.96 | 0.846 | 0.150 | 23.06 | 0.821 | 0.170 |

Table F.2: **Detailed Results on GSO-OOD.** We report the separate OOD evaluation results of the two sets of experiments with input views' maximum elevations $\phi_{\max} = (10°, 20°)$. The average results are used in Tab. 2. Colors indicate the 1st , 2nd best-performing models.

| Methods | $\phi_{max} = 10°$ | | | $\phi_{max} = 10°$ | | | Average | | |
|---|---|---|---|---|---|---|---|---|---|
| | PSNR | SSIM | LPIPS | PSNR | SSIM | LPIPS | PSNR | SSIM | LPIPS |
| MipNeRF360 (Barron et al., 2022) | 22.43 | 0.810 | 0.209 | 23.39 | 0.837 | 0.174 | 22.91 | 0.824 | 0.192 |
| Nerfbusters (Warburg et al., 2023) | 13.84 | 0.633 | 0.329 | 18.06 | 0.724 | 0.269 | 15.95 | 0.678 | 0.299 |
| 2DGS (Huang et al., 2024a) | 22.26 | 0.786 | 0.227 | 24.31 | 0.845 | 0.180 | 23.29 | 0.816 | 0.204 |
| 3DGS (Kerbl et al., 2023) | 20.71 | 0.709 | 0.274 | 22.85 | 0.784 | 0.226 | 21.78 | 0.746 | 0.250 |
| SplatFormer | 24.07 | 0.840 | 0.168 | 25.95 | 0.886 | 0.127 | 25.01 | 0.863 | 0.148 |

$\phi_{\max} = 10°$ and $20°$ respectively. The final evaluation scores reported in Tab. 1 and Tab. 5 are averaged across the two sets of experiments. We also report the separate scores of the two sets in Tab. F.1 (Objaverse-OOD) and Tab. F.2 (GSO-OOD). The evaluation scores of $\phi_{\max} = 20°$ are better than $\phi_{\max} = 10°$ as the input viewing angles are slightly closer to the OOD test views. However in both setups, the elevation deviation, with $\phi_{\mathrm{ood}} \geq 70°$, is quite large and our method outperforms the baselines consistently.

**Visual Comparisons.** We show more visual comparisons with all the baselines in Fig. H.2 (ShapeNet-OOD), Fig. H.4 (Objaverse-OOD), Fig. H.5 (GSO-OOD), and Fig. H.6 (Real-world OOD). We also provide a supplementary video to show the comparisons.

Among standard NVS methods, volumetric representations, such as MipNeRF360 (Barron et al., 2022) and InstantNGP (Müller et al., 2022) often suffer from floater artifacts. While MipNeRF360 excels at capturing fine details in certain examples, its lengthy reconstruction process (7 hours) and slow rendering speed (less than 1 fps) limit its applicability in many real-world tasks.

Among the regularized 3DGS variants, SplatFields (Mihajlovic et al., 2024) produces more regularized Gaussians compared to the standard 3DGS but also loses some fine details in the process. 2DGS (Huang et al., 2024a) generates more surface-aligned Gaussians than 3DGS but still exhibits spiky artifacts due to overfitting to the input views.

For prior-enhanced NVS methods, SSDNeRF (Chen et al., 2023), which is designed to learn category-specific object priors in its original paper, struggles to learn cross-category priors in our training set, resulting in severe artifacts. Nerfbusters (Warburg et al., 2023) incorrectly identifies many structures as floaters, leading to the removal of significant parts of objects. While FSGS (Zhu et al., 2024) achieves a more balanced densification of Gaussians and smoother depth regularization, it only offers marginal visual improvements over 3DGS and continually fails in certain scenarios. InstantSplat (Fan et al., 2024a) also cannot mitigate the artifacts caused by overfitting by using a dense point cloud initialization.

Among learning-based feed-forward methods, LaRa (Chen et al., 2024a) can infer plausible geometry and textures from input views. However, its reliance on a limited number of input views and voxel representation restricts its ability to process and represent high-frequency details. Both SyncDreamer (Liu et al., 2023b) and EscherNet (Kong et al., 2024) suffer from large hallucination errors, producing results that appear visually plausible as single views but are 3D-inconsistent and misaligned with the input views.

# G  BASELINE IMPLEMENTATIONS

**InstantNGP.** We use the officially released code[2] of InstantNGP (Müller et al., 2022). Each scene in our OOD benchmarks is trained for 5k iterations using the default configuration provided in the code. We also tried training the model for more iterations, *e.g*. 20k iterations, but observed no improvement in results. This is most likely due to the fact that we are evaluating on out-of-distribution test camera scenarios. The camera position is scaled and offset, following the paper, to position the reconstructed object within a [0,0,0] to [1,1,1] unit box.

**MipNeRF360.** We use the officially released code[3] of MipNeRF360 (Müller et al., 2022) to produce its results on our OOD benchmarks. The model is trained using batch size 512 for 250k iterations, with learning rate 0.00025. The training of each scene takes approximately 7 hours on a single RTX-4090 GPU. The near and far planes used in volumetric rendering are determined by the ray bounding box intersection following the approach used in InstantNGP. The random background trick used in InstantNGP is also applied to help removing the floaters.

**3D Gaussian Splatting.** We use the gsplat (Ye et al., 2024) and the default hyperparameters of the toolbox. Specifically, we set the number of iterations to 30k, warm-up steps to 500, densifying and culling Gaussians every 500 step, and we stop the density control at 15k steps. For synthetic datasets, we use the ground truth camera poses provided by the Blender rendering. Given that COLMAP struggles to reconstruct certain synthetic objects due to symmetry and smooth textures, we follow the original 3D Gaussian Splatting (3DGS) approach (Kerbl et al., 2023) by randomly sampling 50,000 points within the bounding box of the objects. For our real-world iPhone captures, we first estimate camera poses using all views and then perform point triangulation, with only the input views used to estimate the initial point cloud.

**2D Gaussian Splatting.** We test 2DGS (Huang et al., 2024a) on our benchmarks using the officially released code[4]. The model is trained for 30k iterations using the default configurations, with an additional random background trick used in (Müller et al., 2022) to help removing the floaters.

---

[2]https://github.com/NVlabs/instant-ngp
[3]https://github.com/google-research/multinerf
[4]https://github.com/hbb1/2d-gaussian-splatting

**SplatFields.** We use the officially released code[5] of SplatFields (Mihajlovic et al., 2024). We run the method with its default configuration for over 30k training steps on both ShapeNet and Objaverse datasets.

**InstantSplat.** We use the officially released code[6] of InstantSplat. Assuming known camera poses, we use the provided training and test camera extrinsics and intrinsics, keeping them fixed in both DUST3R (Wang et al., 2024) global alignment and 3DGS optimization. Each of our OOD test scene contains 32 dense training views and applying DUST3R global alignment optimization to all $32 \times 31$ pairs leads to out-of-memory issues and redundant point clouds. Therefore, we use pairwise inference results only for view pairs with overlapping observations, specifically those within an interval of 16 frames. To filter redundant points, we set the depth threshold for InstantSplat's covisibility computation to 0.05 and retain only points with prediction confidence above the 40% quantile for each image. We train 3DGS for 3,000 iterations, using random background and switching off the camera poses optimization. We find that these configurations yield optimal performance. All other settings follow the defaults in the released code.

**FSGS.** We use the officially released code[7] of FSGS (Zhu et al., 2024). We run the method with its default configuration with depth supervision from synthesized psuedo views for 10k training steps on both ShapeNet and Objaverse datasets, and apply random background tricks during training to avoid floater artifacts.

**SSDNeRF.** We use the official code [8] of SSDNeRF (Chen et al., 2023). We train the method using the default hyperparameters for 20k steps on both the ShapeNet and Objaverse datasets, utilizing all available training views. We observe that the proposed image-guided sampling and finetuning of the sampled codes leads to overfitting on the in-distribution training views, and therefore the performance on OOD test views does not exhibit further improvement beyond 20k steps.

**Nerfbusters.** We use the official codebase[9] of Nerfbusters (Warburg et al., 2023). We first train Nerfacto (Tancik et al., 2023) for 30k steps, and then run Nerfbusters on pretrained Nerfacto models for 5k steps to remove the artifacts learned by Nerfacto. The same near and far planes calculation strategy described in MipNeRF360 is employed.

**SyncDreamer.** We use the official codebase[10] of SyncDreamer (Liu et al., 2023b). SyncDreamer supports only single input view, which is insufficient for accurate and faithful out-of-distribution view synthesis. Note that SyncDreamer is computationally demanding—their released model is trained for four days using eight 40GB A100 GPUs, far exceeding our computational budget. For a fair comparison, we train their method on our dataset within the same GPU hours as our approach (three days with eight RTX 4090 GPUs) and use their pretrained checkpoint for initialization.

**EscherNet.** We use EscherNet's released codebase[11]. We train two models on the same ShapeNet-OOD and Objaverse-OOD training sets as our method. Their performance is then evaluated on the corresponding benchmarks. According to the original paper, training EscherNet from scratch requires six A100 GPUs for one week, which exceeds our computational budget. Similar to our approach with SyncDreamer, we initialize training with their released checkpoint pretrained on Objaverse and finetune it on our OOD datasets. During training, we set both the numbers of input and output views to three, using eight RTX 4090 GPUs with a total batch size of eight. For inference, we input all 32 input views to predict all test views at the same time. The models are finetuned for 24k steps, achieving optimal validation performance on the two OOD datasets.

**LaRa.** We use LaRa's released codebase[12]. We train two models on the same ShapeNet-OOD and Objaverse-OOD training sets as our method and evaluate their performances on the respective benchmarks. Due to hardware limitations, LaRa can only process a maximum of four input views on RTX-4090 GPUs. To maximize scene coverage, we randomly select the four most widely sep-

---

[5]https://github.com/markomih/SplatFields
[6]https://github.com/NVlabs/InstantSplat
[7]https://github.com/VITA-Group/FSGS
[8]https://github.com/Lakonik/SSDNeRF
[9]https://github.com/ethanweber/nerfbusters
[10]https://github.com/liuyuan-pal/SyncDreamer
[11]https://github.com/kxhit/EscherNet
[12]https://github.com/autonomousvision/LaRa

| Four of Input Views | SplatFields | MipNeRF360 | 3DGS | Ours | GT |

Figure G.1: **Failure Case.** While our method effectively reduces artifacts in 3DGS (Kerbl et al., 2023) and outperforms SplatFields (Mihajlovic et al., 2024), it does not fully restore some high-frequency details. MipNeRF360 (Barron et al., 2022) excels in detail modeling but suffers from floating issues.

arated input views. For training, we uniformly sample from all available views, incorporating both in-distribution and out-of-distribution views for the 2D image loss. During inference, we feed the model four input views and rasterize the predicted Gaussian primitives to the OOD test views. While it is possible to divide the input views into groups of four, and then combine the processed resulting primitives for rasterization, this approach compromises global consistency and yields worse performance compared to using just four views directly. Following LaRa's default setup (Chen et al., 2024a), we train each model for 150k iterations using 8 RTX 4090 GPUs, which takes approximately 2 days. We initialize training with LaRa's pretrained weights, which speeds up convergence and improves performance.

**Other Baselines.** We acknowledge that some related approaches are not included in this paper due to redundancy or infeasibility. DNGaussian (Li et al., 2024a) supervises 3DGS training with depth maps predicted by monocular depth estimators. The idea is similar to its concurrent work FSGS (Zhu et al., 2024) which, in our OOD-NVS dense capture setup, does not outperform 3DGS. CAT3D (Gao et al., 2024) and ReconFusion (Wu et al., 2024a) use internal image diffusion models for initialization but neither has released code and models. GeNVS (Chan et al., 2023) and 3DiM (Watson et al., 2023) are also publicly unavailable. To compare with methods that utilize pretrained 2D diffusion models, we evaluate SyncDreamer (Liu et al., 2023b) and EscherNet (Kong et al., 2024) which are designed to achieve better multi-view consistency. Other similar approaches like ZeroNVS (Sargent et al., 2024), Zero123 (Liu et al., 2023a), and Vivid123 (Kwak et al., 2024), SV3D (Xie et al., 2024b) only support single-view input. We also tried ViewDiff (Höllein et al., 2024), a similar 2D-prior method where each model is trained for a single object category. However, we found that ViewDiff could not converge when trained on multiple object categories in our cases. Lastly, in the category of learning-based 2D-to-3D methods, SpaRP (Xu et al., 2024), MVSplat (Chen et al., 2024b), and LatentSplat (Wewer et al., 2024) are constrained by the memory limitations of RTX 4090 GPUs, allowing a maximum of four input views. We found that training these methods on our dataset with four large-baseline input views, as we did with LaRa, resulted in failure. This is because these methods rely on overlapping input views to compute cross-view correspondences. We also attempted to partition the consecutive input views into groups and aggregate the predicted Gaussian splats after running each group independently. However, this approach introduced significant global inconsistencies. Consequently, we do not report their results due to their incompatibility with our OOD-NVS task.

## H LIMITATIONS AND FUTURE DIRECTIONS

We outline the current limitations of our method and suggest possible future research directions.

**Fine-grained Details.** Our approach occasionally struggles to recover high-frequency details, particularly in complex textures, as shown in Fig. G.1. While our method reduces artifacts in 3DGS and outperforms previous variants like SplatFields, it still requires improvement in rendering texture details. The limitation may stem from the restricted capacity of our current point transformer backbone which uses grid pooling on the input point cloud to expand the receptive field. Larger grid resolutions and smaller pooling strides can be beneficial, but this requires more computational budgets. Future work can innovate the design of the point transformer architecture, *e.g.* integrating multi-resolution hierarchy, to capture and recover high-frequency details. Designing a trainable adaptive population mechanisim to densify Gaussians in high-frequency regions can also help represent the details.

**Generalization to Real-world Images.** Improving generalization to real-world images remains a key area for future development. Currently, the method is trained exclusively on synthetic datasets, which limits its ability to model complex lighting conditions and textures found in realistic environments. Furthermore, objects in datasets like ShapeNet and Objaverse typically exhibit simpler textures and geometries . To enhance real-world transfer, future work could focus on making the rendering process more realistic and curating larger datasets that include complex 3D assets. Incorporating a balanced mix of synthetic and real-world datasets during training would also help improve the model's performance in more diverse, real-world scenarios.

**Improving Other 3D Representations.** Our approach has the potential to extend beyond 3DGS and be applied to other point-based 3D representations, such as 2DGS (Huang et al., 2024a), which has demonstrated superior performance on OOD-NVS evaluation sets. Training SplatFormer to refine 2DGS could further boost its performance in OOD-NVS tasks. Additionally, incorporating a geometry regularization term for the predicted Gaussian primitives, similar to 2DGS, alongside the photometric loss, presents a promising direction for improving the accuracy and robustness of these representations.

**Diverse Camera Setups.** Our current model focuses on a specific input view distribution where the input views provide full angular coverage along one axis (e.g., azimuth) while having limited angular coverage along another axis (e.g., elevation). The model trained on our current OOD datataset struggles to enhance quality when faced with significantly different input view distributions. For example, in Objaverse-OOD test set, if *high-elevation* views ($\phi \geq 50°$) are used as *input* views and *low-elevation* ($\phi \leq 10°$) views are used as *test* views, our current SplatFormer model provides limited improvement in quality for low-elevation test views. We report the results in Tab. H.1. We attribute this limitation to two factors. First, high-elevation input views capture only a small

Table H.1: **Result of different input view distributions.** For 3DGS *trained* on *high-elevation* views views, our trained SplatFormer achieves only limited improvement in *low-elevation* test views.

|  | PSNR | SSIM | LPIPS |
|---|---|---|---|
| 3DGS | 18.01 | 0.697 | 0.297 |
| SplatFormer (0-shot) | 18.22 | 0.712 | 0.283 |

portion of the scene, leaving much of the lower regions unobserved - areas typically covered by low-elevation views. Since SplatFormer does not learn to generate (or 'hallucinate') completely unseen parts of the scene, it cannot correct artifacts in these unobserved regions. Second, variations in the distribution of input views produce different types of 3DGS artifacts, some of which differ significantly from those encountered during training. We believe this limitation can be addressed by creating a more diverse OOD synthetic training dataset with a wider range of input view trajectories. Equipping point transformer with generative capability and likelihood estimation can be another promising direction.

**Unbounded Scenes.** While this paper primarily focuses on object-centric scenes in this project, the concept of learning data-driven priors for out-of-distribution views holds promise for extending to unbounded, in-the-wild scenes. Since MVImgNet (Yu et al., 2023) serves as a strong testbed, offering substantial real-world multi-view images, we also conduct some preliminary experiments to demonstrate its potential. In

Table H.2: **Results on MVImgNet.**

|  | PSNR | SSIM | LPIPS |
|---|---|---|---|
| 3DGS (Kerbl et al., 2023) | 19.81 | 0.728 | 0.432 |
| SplatFormer | 21.68 | 0.757 | 0.424 |

MVImgNet, each scene is captured via a semi-circular camera trajectory around an object. To study OOD renderings, we split each trajectory into frontal and side views, using one as input and the other as test views. In these OOD test views, 3DGS produces significant artifacts. To adapt our method to this OOD-NVS setting, we constructed a training set with 4k MVImgNet scenes featuring flawed 3DGSs and multi-view images, and trained SplatFormer accordingly. We also create a evaluation set consisting of 70 held-out scenes using the same front-side camera setup. For each evaluation scene, we use the trained SplatFormer to refine the flawed 3DGS, trained only on the frontal/side input views, in a single feed-forward pass. Then we render the refined 3DGS from the side/frontal deviated test views. The evaluation metrics are reported in Tab. H.2, and visual results are shown in Fig. H.1. SplatFormer can remove floaters in the empty space and achieves better metrics than 3DGS. However, it struggles to refine the geometry and appearance of foreground objects. This limitation may stem from the normalization and downpooling operations in the point transformer, which disproportionately downscales foreground objects compared to the large-scale background point clouds, making it difficult for SplatFormer to capture foreground details. We hypothesize that

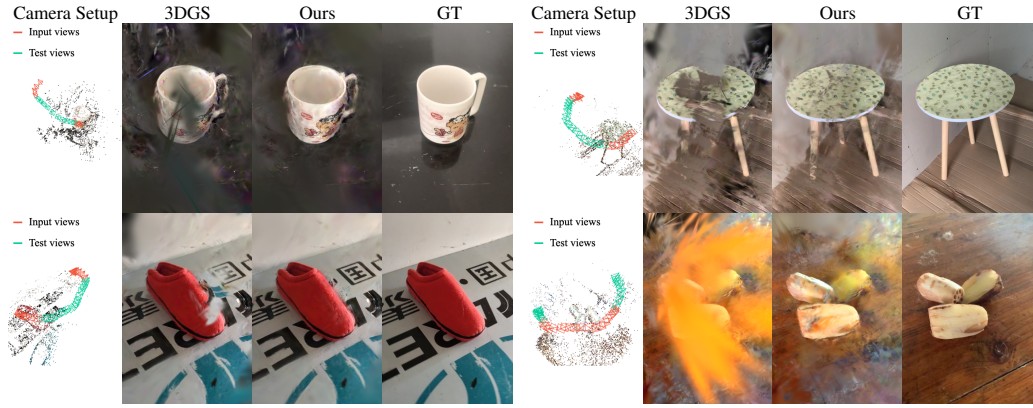

Figure H.1: **The potential of SplatFormer in handling in-the-wild, unbounded scenes.** When trained on MVImgNet (in-the-wild unbounded scenes), SplatFormer learns to partially remove floaters, though refining objects' flawed geometry remains a challenge.

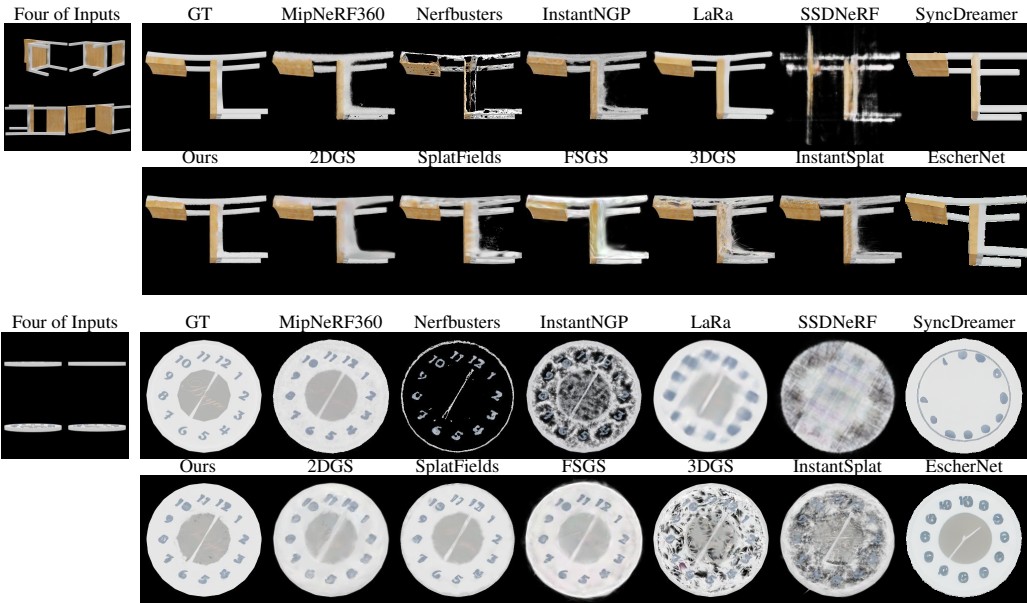

Figure H.2: **Results on ShapeNet-OOD.** We compare our method with baselines: Sync-Dreamer (Liu et al., 2023b), LaRa (Chen et al., 2024a), SSDNeRF (Chen et al., 2023), 3DGS (Kerbl et al., 2023), Nerfbusters (Warburg et al., 2023), SplatFields (Mihajlovic et al., 2024), InstantSplat Fan et al. (2024a), 2DGS (Huang et al., 2024a), FSGS (Zhu et al., 2024), Instant-NGP (Müller et al., 2022), and MipNeRF360 (Barron et al., 2022).

designing a novel adaptive downpooling mechanism within the point transformer may address this issue. Additionally, a divide-and-conquer strategy—*i.e.*, decomposing scenes into objects and background and processing each component separately—could also be beneficial. We plan to explore these directions in future work.

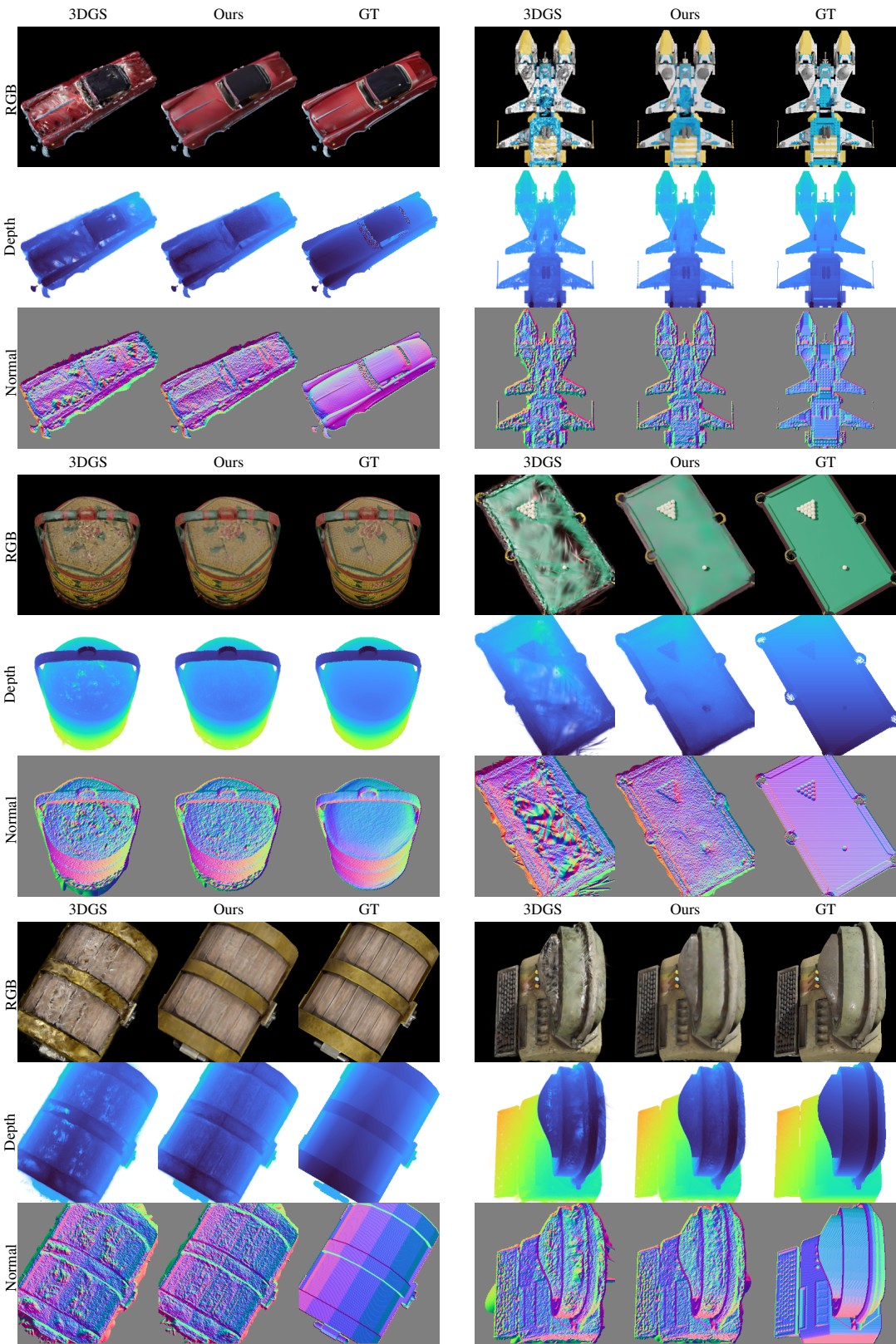

Figure H.3: **Geometry comparison.** While 3DGS is not designed for accurate geometry reconstruction, our method can still enhance the rendered depth and normal of 3DGS.

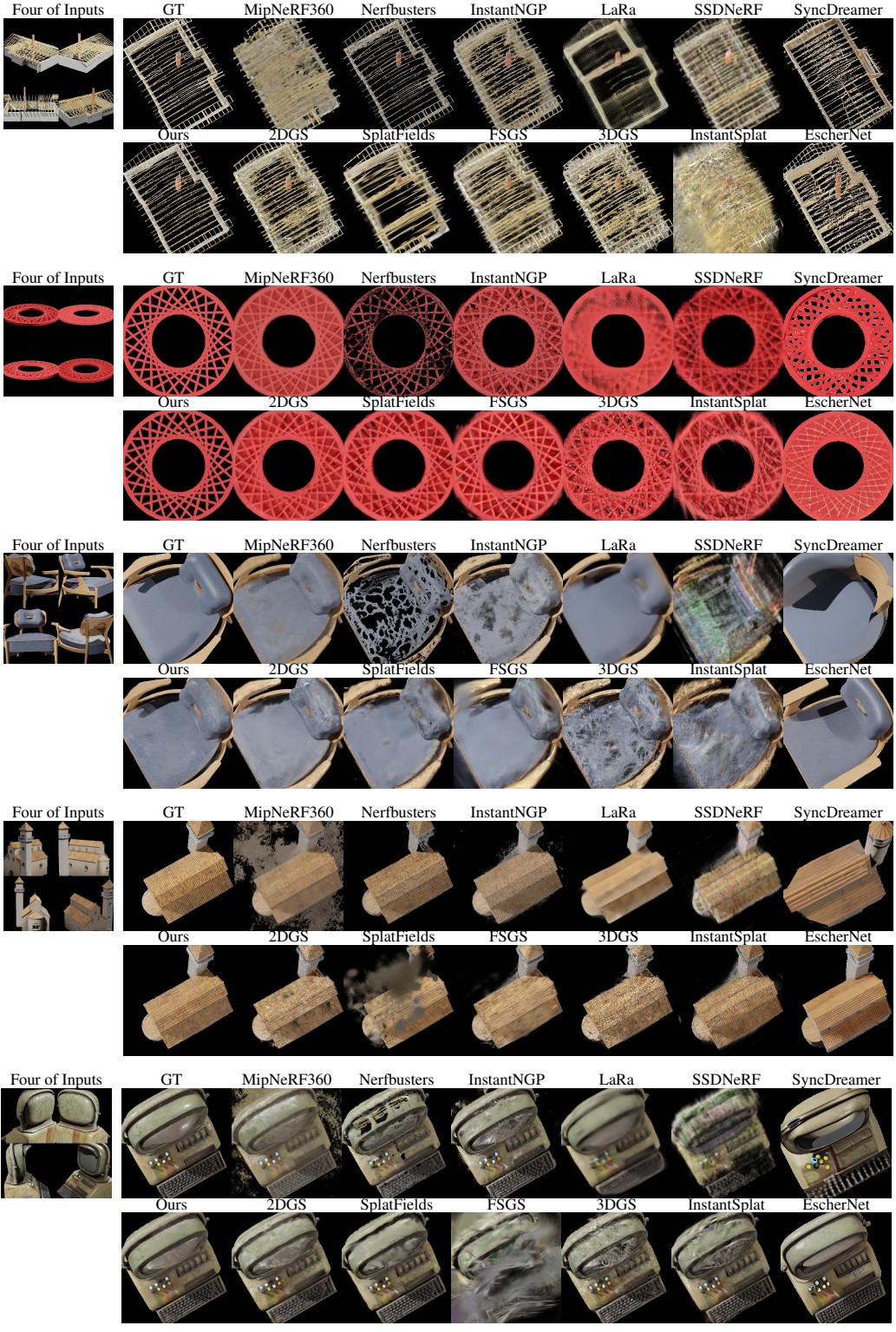

Figure H.4: **Results on Objaverse-OOD.** We compare our method with baselines: Sync-Dreamer (Liu et al., 2023b), EscherNet (Kong et al., 2024), LaRa (Chen et al., 2024a), SSD-NeRF (Chen et al., 2023), 3DGS (Kerbl et al., 2023), Nerfbusters (Warburg et al., 2023), Splat-Fields (Mihajlovic et al., 2024), InstantSplat Fan et al. (2024a), 2DGS (Huang et al., 2024a), FSGS (Zhu et al., 2024), InstantNGP (Müller et al., 2022), and MipNeRF360 (Barron et al., 2022).

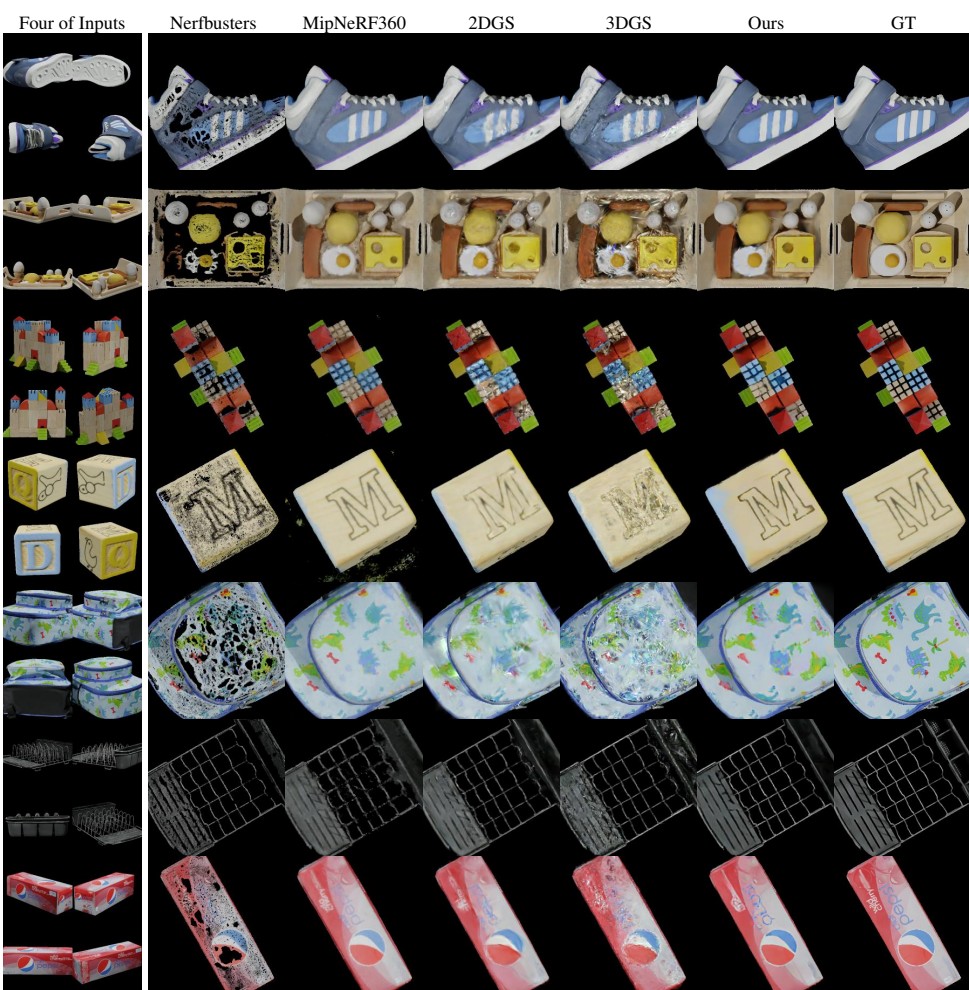

Figure H.5: **Results on GSO-OOD.** We compare SplatFormer, trained on Objaverse scenes, with Nerfbusters (Warburg et al., 2023), 2DGS (Huang et al., 2024a) and MipNeRF360 (Barron et al., 2022).

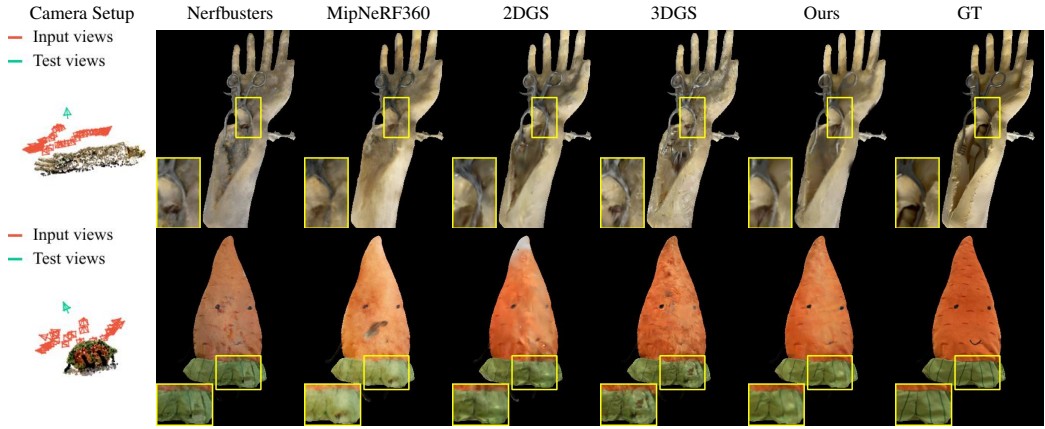

Figure H.6: **Results on Real-World iPhone OOD.** We compare SplatFormer trained on Objaverse with Nerfbusters (Warburg et al., 2023), MipNeRF360 (Barron et al., 2022), 2DGS (Huang et al., 2024a), and 3DGS (Kerbl et al., 2023).

