# OpenReview forum: "SplatFormer: Point Transformer for Robust 3D Gaussian Splatting"
_ICLR.cc/2025/Conference — ICLR 2025 Spotlight_

### Official Review · Reviewer_aVEM · 2024-10-19

**Soundness:** 3
**Presentation:** 3
**Contribution:** 3
**Rating:** 8
**Confidence:** 3

**Summary:**

This paper proposes a method towards enhancing the 3DGS performance on out-of-distribution views. This paper leverages a transformer-based point cloud backbone to encode and refine per-scene optimized 3DGS, and supervise on both interpolated views and out-of-distribution views. The empirical results show that this method leads to signficantly improved quality on OOD views.

**Strengths:**

1. The task of enhancing OOD views quality is important and with good motivation.
2. The idea of training a point cloud backbone to learn priors of OOD views for 3DGS refinement is novel and promising.
3. The experiments are extensive and the results are attractive.
4. The paper is well-written.

**Weaknesses:**

I did not find obvious weaknesses of this paper.

**Questions:**

1. It seems that this method is focused on object-centric scenes with specific camera trajectories (mainly difference in elevations). In both the training datasets and the testing datasets, the input views and OOD views are captured similarly. I'm curious about the results when the trainig views and OOD views are not captured similarly to the training data? For example, if the training views are high-elevation and testing views are low-elevation, or if the training and testing views are with similar elevation but are distant.

---

> ### Author Response · Authors · 2024-11-25
> **Question. Varying input and test views (part 1)**
>
> We thank the reviewer for recognizing the strengths of our submission, including the motivation behind the target task, the novelty and soundness of adapting a 3D point transformer for 3DGS refinement, the comprehensive experimental results, and the clarity of the paper. Below, we address the reviewer’s question concerning the performance of SplatFormer when trained and evaluated with varying input and out-of-distribution test views.
>
> ## **Question. Varying input and test views**
> First, while SplatFormer is trained with supervision only from top-down out-of-distribution (OOD) views with elevations near $90^\circ$ and the input views with elevations $<=15^\circ$, it can refine the input 3DGS during inference to enhance renderings from a wide range of viewpoints across the entire upper hemisphere. In our original submission, Fig. 2 shows that SplatFormer consistently outperforms 3DGS at elevation angles between $20^\circ$ and $90^\circ$. Additionally, SplatFormer improves renderings at varying distances. To demonstrate this, we reduce the camera-to-origin distance by a specified factor and render zoomed-in, close-up views. Below, we report the PSNRs for different elevation angles and camera radii ($R$) on the GSO-OOD dataset.
>
> | Camera Radius \ |Elevation| **$20^\circ$** | **$30^\circ$** | **$40^\circ$** | **$50^\circ$** | **$60^\circ$** | **$70^\circ$** | **$80^\circ$** | **$90^\circ$** |
> |-----------------------------------|-------------------|----------------|----------------|----------------|----------------|----------------|----------------|----------------|----------------|
> | **R = 0.2** | 3DGS               | 22.29            | 21.13           | 20.73           | 19.17            | 18.60            |18.00            | 17.46            | 16.84           |
> |   | Ours       | **22.99**            | **22.19**          | **21.85**            | **21.50**          | **21.44**            | **21.18**           | **20.89**           | **20.18**           |
> | **R = 0.4** | 3DGS               | 23.28           | 22.43           | 21.12            | 19.87            | 18.99           | 18.49           | 18.13            | 17.98            |
> | | Ours     | **23.74**           | **23.14**            | **22.44**           | **21.76**           | **21.38**            | **21.23**            | **21.00**           | **21.04**          |
> | **R = 0.6** | 3DGS               | 25.13           |  23.52           | 22.14           | 21.12           | 20.33            | 19.76            | 19.40          |19.39           |
> |  | Ours      | **25.33**            | **24.04**           | **23.19**           | **22.71**            | **22.43**           | **22.24**           | **22.14**            | **22.22**           |
> | **R = 0.8** | 3DGS               | **27.85**            | 25.86           | 24.07            | 22.74            | 21.89          |21.28           | 20.89           | 20.85           |
> |   |Ours        | 27.28            | **25.87**           | **24.85**           | **24.21**            | **23.89**           | **23.68**           | **23.57**           | **23.61**           |
> | **R = 1.0**|  3DGS               | **29.62**          | 26.65            | 24.58            | 23.30         | 22.50            | 21.97           | 21.70           | 21.66            |
> |               | Ours      | 28.97            | **27.62**            | **26.54**            | **25.77**            | **25.34**           | **25.08**            | **24.93**          |**25.03**           |
>
>
> The results demonstrate that SplatFormer’s refinement improves not only the top-down OOD views used during training but also renderings across a wide range of elevations and camera positions. This is achievable as long as the input captures provide full angular coverage along one axis (e.g., azimuth) while having limited angular coverage along another axis (e.g., elevation).
>
> Notably, when $R \leq 0.6$ and Elevation $\leq 30^\circ$, the test views and input views share similar elevations but differ in their distances to the object. This corresponds to a case mentioned by the reviewer, and in such instances, our trained SplatFormer consistently enhances 3DGS performance.

---

> ### Author Response · Authors · 2024-11-25
> **Question. Varying input and test views (part 2)**
>
> However, we acknowledge a limitation of SplatFormer’s current simplified training procedure, which focuses on a specific input view distribution. It struggles to enhance quality when faced with significantly different input view distributions from those used to optimize the input 3DGS. For example, if only top-down, high-elevation views (Elevation $\geq 50^\circ$) are used for training 3DGS, our current SplatFormer model, trained on the setup with low-elevation views as input views and top-down views as target views, provides limited improvement in quality for low-elevation views (Elevation $\leq 10^\circ$):
>
> | Input view $>=50\degree$, Test View$<=10\degree$  | PSNR |SSIM|LPIPS|
> |---|---|---|---|
> |3DGS|18.01|0.697|0.297|
> |SplatFormer|**18.22**|**0.712**|**0.283**|
>
> We attribute this limitation to two factors. First, high-elevation input views capture only a small portion of the scene, leaving much of the lower regions unobserved - areas typically covered by low-elevation views. Since SplatFormer does not learn to generate (or 'hallucinate') completely unseen parts of the scene, it cannot correct artifacts in these unobserved regions. Second, variations in the distribution of input views produce different types of 3DGS artifacts, some of which differ significantly from those encountered during training. We believe this limitation can be addressed by creating a more diverse OOD synthetic training dataset with a wider range of input view trajectories.
>
> Despite this limitation, the main contribution of our work lies in introducing the important problem of OOD-NVS and proposing a novel architecture and learning paradigm to address it. In future work, we plan to expand and diversify both the training dataset and the applications of SplatFormer, enhancing its robustness to arbitrary shifts in train-test camera distributions and its ability to handle unbounded, large-scale scene processing.

---

### Official Review · Reviewer_MWLJ · 2024-10-27

**Soundness:** 4
**Presentation:** 3
**Contribution:** 3
**Rating:** 8
**Confidence:** 4

**Summary:**

This paper presents SplatFormer, a novel zero-shot model for 3DGS refinement trained on large datasets, in order to enhance the synthesized appearance robustness observed from OOD views. The presented problem OOD-NVS it aims to solve is valuable. Extensive experiments show it achieves SOTA performance on various object-centric datasets.

**Strengths:**

- The presented new problem OOD-NVS is of great value.
- Experiments are extensive, which can well validate the performance of the proposed method.
-  SplatFormer achieves SOTA performance on various object-centric datasets in OOD-NVS task compared to current related methods.

**Weaknesses:**

- Although some experiments using real-world datasets are conducted, all involved datasets are still mainly object-centric. It is still a problem that if this learning-based method can be applied to real-world and non-object-centric scenes with more complex foreground and background. The corresponding data are much more difficult to collect than the object-centric data, and also more difficult to process and use in training.
- Lack of reporting geometry results. Although there are many comparisons in appearance, it's another important problem that how much can the refinement benefit the reconstructed geometry. However, there are no results like depth and surface normal are shown.

**Questions:**

- Would like to see some discussion and exploration for non-object-centric scenes.
- Would like to see more comparisons on geometry, like surface normal and depth.

---

> ### Author Response · Authors · 2024-11-25
> **Weakness 1/2. Applicability to Non-Object-Centric Scenes**
>
> We thank the reviewer for their thoughtful feedback and for recognizing the strengths of our work, including the importance of addressing the OOD-NVS problem, the extensive experiments validating our approach, and the state-of-the-art performance achieved on various object-centric datasets.
>
> Below, we address the reviewer’s discussion points: (W1) **Non-Object-Centric Scenes**: Exploring the applicability of our method to real-world, non-object-centric scenes with more complex foregrounds and backgrounds; (W2) **Geometry Comparisons**: Providing additional comparisons on reconstructed geometry, including metrics such as depth and surface normals. We hope these responses address the reviewer’s questions and further clarify the contributions of our work.
>
> ## **Weakness 1/2. Applicability to Non-Object-Centric Scenes**
>
> We appreciate the reviewer’s recognition of our experiments on real-world objects. We believe that strong performance in object-centric scenes can contribute to addressing complex scenes, particularly if a compositional approach is used to decompose the foreground and background, allowing SplatFormer to process individual objects separately.
>
> Additionally, we would like to direct the reviewer’s attention to Section F, "Limitations and Future Directions" (**Lines 1162-1185**) of the Appendix, where we discussed SplatFormer’s potential for real-world unbounded scenes. Specifically, we conducted an experiment using the MVImgNet [5] dataset, a large-scale collection of multi-view images. For each capture in MVImgNet, we split the views into frontal views and side views, treating one as input views and the other as OOD test views. We trained SplatFormer on a curated MVImgNet-OOD dataset and tested it on OOD views of held-out scenes. More experimental details are provided in Appendix F. For reference, we include the test results from Table F.1 below:
>
> | Method |PSNR |SSIM|LPIPS|
> |---|---|---|---|
> |3DGS [8] |19.81|0.728|0.432|
> |SplatFormer|**21.68**|**0.757**|**0.424**|
>
> We provided visual results in Figure F.1 and added additional results in Figure G.2 of the appendix in our updated manuscript, demonstrating that SplatFormer reduces floater artifacts and improves geometry in certain cases. However, we acknowledge its limitations in enhancing high-frequency details, which may be due to the need for a larger-capacity point transformer or a novel multi-scale hierarchical design to handle large-scale, irregularly spaced point clouds.
>
> Regarding the challenge of acquiring real-world datasets, we speculate that incorporating a diverse range of synthetic datasets could reduce reliance on massive-scale real-world datasets. Notably, training our model exclusively with synthetic data has already demonstrated promising results on real-world test scenes (Table 2, Figure 5, and Figure F.5).
>
> Finally, unbounded scene reconstruction from limited observations remains an open challenge, with most existing prior-enhanced NVS methods [1, 2, 3, 4] primarily focused on object-centric scenes. In contrast, SplatFormer not only excels in these settings but also demonstrates promising potential for unbounded scenes. Enhancing the network architecture and incorporating more synthetic and real-world training data will be key areas of focus in our future work.

---

> ### Author Response · Authors · 2024-11-25
> **Weakness 2/2. Geometry results**
>
> We thank the reviewer for raising this valuable question. We will include a geometry comparison experiment in our revised manuscript, as presented in Figure G.3 of the updated appendix.
>
> To quantitatively evaluate performance, we measure the mean absolute error (MAE) between the ground-truth depth and normal maps, and the corresponding rendered depth and normal maps, under out-of-distribution (OOD) views for 3DGS [8] and our method. Specifically, the depth maps for 3DGS and our method are rendered as the weighted average depth of Gaussian primitives, which is a common way to derive depth maps from 3DGS as used in the gsplat toolbox [9] and other 3DGS-related work [10, 16]. We then follow 2DGS [17] to compute the normal maps using finite differences on the estimated surface derived from the depth maps.
>
> The results show that, in addition to improving rendering quality, our method enhances the accuracy of rendered depth and normal maps:
>
> | Results on Objaverse-OOD | Depth-MAE (x1e-4) | Normal-MAE |
> |---|---|---|
> |3DGS [8] |6.70|0.239|
> |SplatFormer|**4.05**|**0.214**|
>
> We also provide a visualization of the rendered depth map in Figure G.3 of the updated appendix. In our revised manuscript, we will include more comprehensive qualitative and quantitative results on geometry.
>
> Additionally, since 2DGS [17] produces more regularized depth and normal maps than 3DGS, our method could potentially further improve geometry results when applied to 2DGS refinement, as discussed in Lines 1128–1133 of the Appendix.

---

> > ### Comment · Reviewer_MWLJ · 2024-11-25
> >
> > Thanks for the detailed responses from the authors. My concerns have been well solved. The promoted OOD problem is worthy of further research, and the method's generalizability is impressive, showing promising results in real-world data. I'd like to raise my rating to 8, despite some challenges still existing for its application to more real-world applications.

---

### Official Review · Reviewer_MA5d · 2024-10-30

**Soundness:** 3
**Presentation:** 3
**Contribution:** 3
**Rating:** 8
**Confidence:** 3

**Summary:**

Although existing 3D representations such as 3DGS or Nerf can achieve novel view synthesis, their rendering performances on OOD views with relatively large elevations are relatively limited. This may come from the large differences between training and evaluation OOD views. In this work, the authors propose a framework, named SplatFormer, using transformer to refine the optimized 3DGS for better performances under OOD views. Benefited from the training under both normal and OOD views, SplatFormer can indeed improve the rendering under OOD views.

**Strengths:**

1. The core contribution of this work to refine 3D Gaussians with a genelizable transformer is meaningful;
2. The authors construct training and evaluation sets for the claimed OOD problem, from ShapeNet and Objaverse dataset.
3. Extensive experiments with different baselines on multiple datasets confirm that the proposed method can obviously improve the rendering performances under poses with large elevations.

**Weaknesses:**

My major concerns lie on that some comparisons between the proposed method and baselines  may be not so fair. For example, the optimization of the proposed method use 32 low-elevation views, while the results of some methods, e.g., LaRa, take only 4 views for input. The lack of training views may naturally affect its performances. Can we apply the proposed framework to the 3D Gaussian primitives generated by LaRa directly? In this way, the performances of proposed refinement might be evaluated more fairly.

**Questions:**

Except the mentioned problem in the weakness section, I have some other problems.
 1. As the method is mainly proposed to address the problems of rendering under relatively large elevations, the limitation of performances may come from the lack of corresponding training views. Could we just use some novel view synthesis baselines, such as Zero123, SV3D to generate pseudo images from such poses with large elevations, and then optimize 2DGS, 3DGS, etc. for reconstruction? Would this also improve the performances under poses with large elevations?
 2. What is the specific settings for the training of the Gaussian primitive transformer? Would it select input views and OOD views randomly? Does different selection strategies have influences on the final performances?
 3. How is the efficiency of the transformer? As the density of Gaussian primitives might be quite high after optimization, wouldn't it take great time and memory cost to incorporate such a transformer framework?

**Details Of Ethics Concerns:**

NA.

---

> ### Author Response · Authors · 2024-11-25
> **Weakness. Additional Comparison with LaRa on a 4-view Setup**
>
> We thank the reviewer for their thoughtful feedback and for recognizing the strengths of our work, including the meaningful contribution of a generalizable transformer for refining 3D Gaussian splats, the construction of datasets tailored to the OOD problem, and the extensive experiments validating the effectiveness of our method.
>
> In the following, we address the identified weakness (W1) concerning comparisons with LaRa under unfavorable conditions and respond to three discussion points: (Q1) utilizing pseudo-OOD views for view synthesis, (Q2) the training setup and OOD view selection, and (Q3) the analysis of SplatFormer’s resource efficiency.
>
> The reference list for the cited papers is provided in our general response.
>
> ### **Weakness. Additional Comparison with LaRa on a 4-View Setup**
>
> We appreciate the reviewer’s comment and suggestion. Since LaRa is computationally expensive and limited to a maximum of four input views due to memory constraints on standard GPUs (e.g., RTX-4090, 24 GB), we can only provide four input views to LaRa for our OOD-NVS tasks, which involve 32 input views. However, following the reviewer’s suggestion, we are happy to perform an additional comparison under the four-view setup by applying our framework to the Gaussian splats generated by LaRa [4].
>
> #### **Experiment Details:**
> We used LaRa's released checkpoint to predict 2DGS from _four input views_ for our Objaverse training scenes. Subsequently, we trained our SplatFormer to refine LaRa’s outputs using the same procedure described in the experiments section of our paper. Even in this constrained four-view setup, SplatFormer achieved a noticeable improvement over LaRa’s baseline performance on the Objaverse-OOD and GSO-OOD test sets.
>
> |            Four input views               | **Objaverse-OOD** | **GSO-OOD**  |
> |---------------------|------------------------------|------------------------------|
> |          | PSNR / SSIM / LPIPS | PSNR / SSIM / LPIPS |
> | LaRa [4]              | 16.87 / 0.640 / 0.352       | 17.91 / 0.677 / 0.339       |
> | LaRa + SplatFormer | **18.29 / 0.688 / 0.275**       | **18.83 / 0.714 / 0.279**       |
>
> #### **Discussion:**
> It is important to note that LaRa’s 2DGS encodes information from only four input views, which often results in suboptimal visual quality with significant blur artifacts, particularly when viewed from OOD perspectives. While SplatFormer enhances these outputs, fully reconstructing OOD views from such an extremely sparse input-view setup remains highly challenging. This limitation arises from the inherently ill-posed nature of the reconstruction task under these conditions.
>
> Our method, by contrast, is specifically designed for scenarios with denser input views, where more comprehensive information is available for reconstructing high-fidelity outputs. This design aligns with practical applications, such as immersive navigation or surgical visualization (L:048), where capturing dense, calibrated views is both feasible and necessary. Accordingly, in our main paper (Table 1), we focus on comparisons in the dense-input setup, emphasizing LaRa’s limitations in scaling to scenarios with dense input views.

---

> ### Author Response · Authors · 2024-11-25
> **Question 1/3. Pseudo-OOD Views for 3DGS training**
>
> We appreciate the reviewer’s insightful suggestion. While leveraging pseudo-OOD views from diffusion-based novel view synthesis (NVS) baselines is an interesting approach, we observed that such views often lack plausibility and introduce artifacts. Incorporating these erroneous pseudo views into the 3DGS training pipeline can actually degrade the final reconstruction quality compared to 3DGS [8].
>
> #### **Experiment Details:**
>
> To illustrate this, we conducted experiments using state-of-the-art open-source diffusion-based NVS models: SV3D [2] and EscherNet [3]. As detailed in our response to Reviewer K5mf, EscherNet outperformed other baselines, including SyncDreamer [1] and SV3D [2], due to its ability to process multiple input views, whereas other diffusion-based methods are limited to a single input view. By combining the pseudo-OOD views generated by SV3D or EscherNet with the ground truth input views, we trained 3DGS and obtained the following results:
>
> | **Method**         | **PSNR** | **SSIM** | **LPIPS** |
> |---------------------|----------|----------|-----------|
> | SV3D [2] (0-shot) | 10.93  | 0.498 | 0.455 |
> | SV3D [2] (0-shot+Distill)  | 14.19 | 0.562  | 0.405 |
> |  EscherNet [3] (Finetune) | 16.57 | 0.633  | 0.273 |
> | EscherNet [3] (Finetune+Distill)| 18.88 | 0.701  | 0.258 |
> | 3DGS               | 21.78   | 0.746    | 0.250     |
> | Ours               | **25.01** | **0.863** | **0.148** |
>
> A visual comparison is included in Figure G.1 and the second uploaded video in the supplementary material (file name: _compare_with_diffusion-sparse-baselines.mp4_).
>
> #### **Discussion:**
> Our results indicate that pseudo-OOD views generated by diffusion-based methods are often suboptimal, introducing errors that propagate through the training pipeline and degrade performance. This is reflected in the lower PSNR, SSIM, and LPIPS values for models trained with pseudo views compared to our method.
>
> In contrast, our approach directly refines 3DGS representations using a point transformer, enabling robust generalization to challenging OOD poses without relying on pseudo views. This avoids the pitfalls of hallucinated artifacts and ensures that reconstructions maintain high fidelity.
>
> Further discussion on the limitations of sparse-view and diffusion-based NVS methods is provided in our response to Reviewer K5mf ("Weakness 1: Generative Diffusion Priors for OOD-NVS").

---

> ### Author Response · Authors · 2024-11-25
> **Question 2/3. SplatFormer's Training Setup**
>
> We thank the reviewer for raising this insightful question. In the revised manuscript, we will include an ablation study to analyze the impact of the ratio of OOD views to input views during SplatFormer’s training on the final model performance.
>
> As noted in Lines 860–861 of Appendix D in our manuscript, our training setup involves rendering four target images per scene, with a default setting of 70% OOD views and 30% input views.
>
> #### **Experiment Details**
> To investigate the effect of varying the OOD view ratio, we conducted an ablation study. Specifically, we trained four SplatFormers on the Objaverse-OOD training dataset, using different ratios of OOD views and input views for the photometric loss. We then evaluated the models' performance on various test views. Then we evaluate the models’ performance on different test views.  Due to computational constraints, this study was performed using a fraction of the computational resources allocated for our full model (4x RTX-4090 GPUs with  200k steps and the gradient accumulation step set to 2).
>
> Metrics were evaluated on Objaverse-OOD test scenes across three elevation ranges: **low** (20°–30°), **mid** (40°–60°), and **high** (70°–90°).
>
> |        | **Low (20°–30°)**       | **Mid (40°–60°)**       | **High (70°–90°)**      |
> |----------------------|-------------------------|-------------------------|-------------------------|
> |                      | **PSNR / SSIM / LPIPS**| **PSNR / SSIM / LPIPS**| **PSNR / SSIM / LPIPS** |
> | **90% OOD**         | 25.07 / 0.883 / 0.119  | 23.04 / 0.826 / **0.160**  | **22.85 / 0.818 / 0.172** |
> | **70% OOD (default)**| 25.61 / 0.892 / 0.111  | 23.00 / **0.832** / 0.161  | 22.60 / 0.810 / 0.179  |
> | **50% OOD**         | 25.99 / 0.896 / 0.108 | **23.09** / 0.827 / 0.165 | 22.53 / 0.803 / 0.187  |
> | **30% OOD**         | **26.22 / 0.897 / 0.108**  | 22.82 / 0.816 / 0.177  | 21.88 / 0.784 / 0.207  |
> | **3DGS**            | 25.80 / 0.873 / 0.132  | 20.92 / 0.736 / 0.237  | 19.24 / 0.673 / 0.285  |
>
>
>
> #### **Discussion**
> Our results indicate that increasing the ratio of OOD views during training improves performance at extreme viewpoints (e.g., high elevations), though it slightly compromises rendering fidelity at lower elevations. We selected the default 70% OOD ratio as it provides a balance between enhancing OOD views and preserving high fidelity for in-distribution views.
>
> We plan to complete this study using the full computational budget and include it in the revised manuscript to provide a more comprehensive analysis of the impact of OOD and input view ratios.

---

> ### Author Response · Authors · 2024-11-25
> **Question 3/3. The Resource Efficiency of SplatFormer**
>
> We thank the reviewer for their insightful question regarding the efficiency of SplatFormer. In the final manuscript, we will highlight SplatFormer's resource efficiency and include a detailed analysis and comparison to emphasize its scalability and practicality.
>
> #### **Memory and Inference Time Analysis**
> To assess SplatFormer's efficiency, we measured its memory usage and inference time across varying numbers of Gaussian splats. The results are summarized as follows:
> | **Number of Splats:**       | 10k-40k |  60k-70k  |  90k-100k  |  _Average_ |
> |---------------------|-------------|-------------|--------------|-------------|
> | Memory Usage (MB): | 675         | 801         | 1010         | 782         |
> | Inference Time (ms):     | 62          | 70          | 118          | 75          |
>
> In the GSO test set, the average number of input splats is 50k, with 80% of inputs containing fewer than 70k splats - an amount sufficient to capture high-frequency details. These results demonstrate that SplatFormer scales efficiently with increasing numbers of Gaussian splats, with memory usage and inference time remaining within practical bounds.
>
> #### **Comparison with Other NVS Methods**
> To provide additional context, we benchmarked SplatFormer's performance against other state-of-the-art NVS methods, including the diffusion-based EscherNet [3] and LaRa [4], which specifically highlights efficiency as a key advantage. The results are as follows:
>
> | **Method**     | **Encoder Type**   | **Output** | **Params (MB)** | **Inference time (ms)** | **Memory Usage (MB)** |
> |-----------------|-----------------------|------------------------|------------------|-------------------|------------------------|
> | **EscherNet** [3]   | SVD [2]     |  Single img   | 971              | 1402             | 2178                  |
> | **LaRa** [4]       | DINO [14]        |   GS splats        | 125              | 157              | 1508                  |
> | **Ours** | 3D Point Transformer [15]  | GS splats | **48**           | **75**           | **782**               |
>
>
> We measured LaRa’s inference time and memory usage for generating 2D Gaussian splats (2DGS) from four input views, and EscherNet’s for producing a single output view using the same four input views.
>
> Notably, another advantage of our method is that it generates 3DGS splats in a fast single pass, enabling real-time rendering into any view. This is significantly more efficient than approaches like SV3D [2] and EscherNet [3], which rely on 2D diffusion generative models and require multiple time-consuming diffusion steps to produce only a single frame.
>
>
> The results demonstrate that SplatFormer achieves significantly better inference time and memory efficiency while utilizing a more lightweight architecture, making it a more scalable model.
>
> We will include a discussion on efficiency in the revised manuscript.

---

> > ### Comment · Reviewer_MA5d · 2024-11-26
> > **Response to the authors**
> >
> > Thanks for the authors' detailed response. The additional experiments have effectively addressed my concerns, particularly the fairer comparisons with LaRA, which confirm that the proposed method provides valuable refinements for existing GS-based generation approaches. While the application on object-level data remains somewhat limited, considering the extensive experiments and well-founded motivation, I will raise my score.

---

### Official Review · Reviewer_K5mf · 2024-11-04

**Soundness:** 2
**Presentation:** 3
**Contribution:** 3
**Rating:** 6
**Confidence:** 5

**Summary:**

The paper introduces SplatFormer, a point transformer model for refining 3D Gaussian Splatting (3DGS) representations under out-of-distribution (OOD) view conditions (with initialized Gaussian Splats). This is motivated by 3DGS struggles with quality degradation when test views differ significantly from training views. SplatFormer addresses this by learning to refine Gaussian splats, leveraging attention mechanisms to maintain consistency across viewpoints and removing artifacts in OOD scenariosl, with the collected large-scale object-centric data. The approach outperforms prior methods in robustness on several test datasets.

**Strengths:**

- The paper introduces a novel and important direction for rendering at unseen, highly relevant test views, addressing a significant gap in current 3D rendering research.
- By employing point transformers for aggregating Gaussian splats, the method offers a sound and efficient approach to achieve improved detail and visual fidelity.

**Weaknesses:**

- The paper does not explore the potential of utilizing generative priors for OOD-NVS, particularly by introducing diffusion models to assist in hallucinating unseen views, which could enhance performance in novel view synthesis in a more reasonable way.
- The study is primarily focused on object-centric cases, despite the availability of scene-level 3D datasets (scannet, scannet++, blendedmvs, megascene, megadepth, mvsimgnet). Expanding the scope to scene-wise data could provide a broader basis for extrapolation and robustness in more complex environments.
- For object-centric cases, single-image-to-3D methods may suffice for preserving geometric consistency and hallucinating texture details. It is unclear why some introduced baselines, including generalizable GS and sparse-view GS, underperform in these scenarios relative to expectations.

**Questions:**

Please refer to the questions in the weaknesses section concerning the problem-solving approach and dataset scope. The reviewer strongly suggests that the authors include a video comparison, as novel view synthesis is highly dependent on visual assessment.



--------------------
Thank you for the detailed explanation and for addressing my concerns. After reviewing the comments from other reviewers and considering your explanation regarding the broader scope of your work, I agree that the merit of addressing OOD challenges in neural rendering using a large-scale model is valuable.

I appreciate the clarification and the balance you’ve struck in presenting the scope of your contributions, the promised clarification in the future abstract and introduction. As a result, I will raise my score to support the acceptance of your submission.

---

> ### Author Response · Authors · 2024-11-25
> **Weakness 1/3.  Generative Diffusion Priors for OOD-NVS  (part-1)**
>
> We thank the reviewer for their valuable feedback and for recognizing several positive aspects of our submission, including addressing a significant challenge in neural rendering and proposing an efficient and well-founded approach using Point Transformer 3DGS refinement.
>
> Below, we address the main discussion points raised by the reviewer: (W1) the utilization of generative priors for out-of-distribution novel view synthesis tasks, (W2) the method's capability for scene-level reconstruction and handling unbounded scenes, (W3) the reasons behind the underperformance of single-image-to-3D models, and (Q1) the request for video comparisons.
>
> ### **Weakness 1/3. Generative Diffusion Priors for OOD-NVS (part-1)**
>
> Generative models, particularly diffusion models, hold significant potential for improving novel view synthesis. We provide a discussion of these methods in the original submission (lines 184-186) and perform a comprehensive quantitative evaluation in the OOD-NVS setup against representative baselines, including SyncDreamer [1], SSDNeRF [12] (Table 1, Figure F.2, Figure F.3), and DiffBIR [13] (Table 3, Figure C.1).  The selected methods encompass diverse approaches to integrating diffusion-based priors, including single-image-to-3D techniques leveraging 2D diffusion priors (SyncDreamer), methods utilizing diffusion-based 3D priors (SSDNeRF), and pipelines focused on diffusion-based image enhancement (DiffBIR).
>
> Our analysis highlights the following limitations of these methods within the considered OOD-NVS scenario:
> * **Erroneous Hallucination under OOD Views**: Generative models may hallucinate content in OOD views that is absent from the input views. In real-world scenarios requiring accurate reconstruction, such as surgical scene reconstruction, this behavior presents significant challenges to the models' applicability.
> * **Limited Capacity for Handling Multiple Input Views**: Most models are restricted in the number of input views they can effectively process and fail to leverage the dense image sets that are often available in the considered OOD-NVS scenario.
> * **Multi-view Inconsistency and Flickering Artifacts**: These methods struggle to maintain temporal and spatial coherence across generated views.
>
> To further validate our findings, we evaluated additional state-of-the-art open-source diffusion-based methods and extended the comparison to include some previously discussed approaches. The evaluated methods are as follows:
>
> * SyncDreamer (ICLR 2024) [1]: A single-image-to-3D method leveraging Stable Diffusion [6]. Please note that the method was already evaluated in the original submission.
> * SV3D (ECCV 2024) [2]: A single-to-3D method utilizing Stable Video Diffusion [7].
> * EscherNet (CVPR 2024) [3]: To the best of our knowledge, the only open-source diffusion-based method capable of processing more than 10 input views.
>
> For SyncDreamer and SV3D, we use the first frame of the input view set as the input condition. For EscherNet, we use all of the 32 input views as input conditions.
>
> For a fair comparison with our model, we fine-tuned the released checkpoints of SyncDreamer and EscherNet using the same Objaverse-OOD training set and computational resources employed for SplatFormer. Fine-tuning was necessary for SyncDreamer, as the released model is limited to generating its predefined 16 novel views, and it also enhanced EscherNet’s performance on our OOD test set. Since the concurrent work SV3D does not provide its training script, we tested its released checkpoint in a zero-shot manner.
>
> To address the multi-view inconsistency in the per-frame generated results of these diffusion models, we also attempted to optimize 3DGS using the output of the diffusion models. Specifically, we employed the fine-tuned EscherNet and the released SV3D to generate 9 OOD views. These generated novel views, combined with 32 ground-truth input views, were used to train a 3DGS. We then evaluated the OOD renderings produced by the distilled 3DGS, denoting these experiments as EscherNet$\rightarrow$3DGS and SV3D$\rightarrow$3DGS. Notably, these experiments also directly address a question raised by Reviewer MA5d (Question 1/3: Pseudo-OOD Views for 3DGS training).
>
> Please refer to the following sections for the experimental results.

---

> ### Author Response · Authors · 2024-11-25
> **Weakness 1/3. Generative Diffusion Priors for OOD-NVS (part-2)**
>
> We report metrics on the **GSO-OOD** dataset:
> | Results on GSO-OOD | PSNR  | SSIM  | LPIPS   |
> |---|---|---|---|
> | SyncDreamer [1] (finetune) | 11.86 | 0.518  | 0.451 |
> | SV3D [2] (0-shot) | 10.93  | 0.498 | 0.455 |
> | SV3D$\rightarrow$3DGS | 14.19 | 0.562  | 0.405 |
> |  EscherNet [3] (0-shot) |  13.74 | 0.585  | 0.367|
> |  EscherNet (finetune) | 16.57 | 0.633  | 0.273 |
> |  EscherNet$\rightarrow$3DGS | 18.88 | 0.701  | 0.258 |
> | 3DGS [8] |   21.78  | 0.746  | 0.250 |
> | Ours |   **25.01**  | **0.863**  | **0.148** |
>
>
> We have included a video comparison in our updated supplementary material (file name: _compare_with_diffusion-sparse-baselines.mp4_). We encourage the reviewer to watch the video, as it visually demonstrates the limitations of diffusion-based methods.
> #### **The following observations can be made from the presented quantitative and qualitative results:**
> * Although diffusion-based baselines generally produce visually plausible OOD views, they often hallucinate scene elements that are inconsistent with the input views. This issue also affects EscherNet, even when provided with all 32 input views. For example, when processing Adidas (c) shoes with the iconic three parallel stripes, EscherNet mistakenly hallucinates a fourth stripe in its OOD view.
> * Likewise, the hallucination error propagates into the distilled 3DGS, making it incapable of addressing the artifacts in OOD views.
> * These issues are also reflected in the quantitative results of all the baselines, which show a PSNR more than 6 points lower than our method (EscherNet$\rightarrow$3DGS: 18.88 vs. Ours: 25.01).
>
> In contrast, our method addresses the limitations of diffusion-based NVS through holistic reasoning over a set of 3DGS primitives. Another key advantage of our approach lies in its **computational efficiency**: the input splats are refined in a single pass through the 3D point transformer, eliminating the need for the computationally expensive denoising process required to generate each image in diffusion-based approaches.
>
> Nonetheless, we recognize the impressive capability of diffusion-based methods to generate complex, photorealistic scenes with minimal input. Exploring ways to combine their strengths with the 3D consistency, efficiency, and robustness of our method offers a compelling direction for future research.

---

> ### Author Response · Authors · 2024-11-25
> **Weakness 2/3, 3/3 and Question on the video**
>
> ###  **Weakness 2/3. Scene-level Refinement, Unbounded Scenes**:
>
> Recognizing the importance of addressing OOD-NVS in complex scenes, we discussed SplatFormer’s potential for real-world unbounded scenarios in Section F: Limitations and Future Directions (**Lines 1162-1185**) of the originally submitted Appendix. To evaluate its performance, we applied our proposed framework to the MVImgNet dataset [5]. On the test set, our method outperforms 3DGS:
>
> | Method |PSNR |SSIM|LPIPS|
> |---|---|---|---|
> |3DGS [8] |19.81|0.728|0.432|
> |SplatFormer|**21.68**|**0.757**|**0.424**|
>
> We present the visual comparison in Figure F.1 and Figure G.2, showing that SplatFormer reduces floater artifacts and improves geometry in many cases. However, we acknowledge its limitations in enhancing high-frequency details, likely due to the current network's insufficient capacity for processing large-scale, irregularly distributed point clouds. Future improvements could involve designing a novel multi-scale hierarchical point transformer architecture to handle large scenes and incorporating real-world training data alongside synthetic data. Additionally, since SplatFormer demonstrates strong generalization on real-world objects (Table 2, Figure 5, and Figure F.5), it may be feasible to decompose the scene and process individual objects separately.
>
> Unbounded scene reconstruction from limited observations remains an open challenge, with a majority of existing prior-enhanced NVS methods [1,2,3,4] also focusing on object-centric scenes. In contrast, SplatFormer not only excels in such settings but also shows promising potential for unbounded scenes. Improving the training strategy and network architecture will be our focus in future work.
>
> ###  **Weakness 3/3. The Underperformance of Sparse-view Reconstruction Methods**:
>
> In our earlier response (Weakness 1), we addressed state-of-the-art open-source diffusion-based single-image-to-3D models, SyncDreamer [1] and SV3D [2]. Evidence shows that relying on a single input view introduces significant ambiguity, leading the model to hallucinate novel views that are misaligned with the input capture.
>
> To further extend the discussion, we compare our approach to LaRa [4], a state-of-the-art method that predicts 2D Gaussian splats from four input views. While LaRa achieves impressive results in its original paper, we demonstrate that it struggles to produce high-quality OOD views.
>
> In the original submission, we evaluated a fine-tuned version of the LaRa framework on the Objaverse-OOD and ShapeNet-OOD datasets. Here, we extend this evaluation by testing two LaRa models on the GSO-OOD test set: the released checkpoint model and the model fine-tuned on our Objaverse-OOD dataset.  To highlight LaRa's degradation in OOD views, we report metrics in both in-distribution views (elevation<=10$\degree$) and out-of-distribution views (elevation>=70$\degree$).
>
> | Result on GSO |  In-distribution Views | Out-of-distribution Views |
> |---|---|---|
> |  |  PSNR/SSIM/LPIPS| PSNR/SSIM/LPIPS |
> | LaRa [4] (0-Shot$^\dagger$) | 24.39/0.870 /0.158 |  17.91/0.677/ 0.339|
> | LaRa [4] (Finetune)  |25.22/0.880/0.152  |  19.87/0.721/0.310|
> |  Ours |**30.55/0.961/0.057**   | **25.01/0.863/0.148**|
>
> $\dagger$: LaRa reports a PSNR of 29.15 on the GSO test set used in their paper. However, we observe that their test images primarily feature empty backgrounds, which inflates the PSNR score. In contrast, our test set includes extensive foreground regions, presenting a more challenging evaluation scenario.
>
> We include a visual comparison between the fine-tuned LaRa and ours in Figure G.1, and the two supplementary videos (_compare_with_major-baselines.mp4_,  _compare_with_diffusion-sparse-baselines.mp4_), demonstrating that LaRa's outputs suffer from noticeable blurriness and performance degradation in OOD views.
>
> The results of SyncDreamer [1], SV3D [2], and LaRa [4] indicate that existing single/sparse-to-3D methods struggle to produce accurate OOD views due to their inability to utilize more than four input views to resolve ambiguity. In contrast, our method is agnostic to the number of input views, as it operates directly on the initial 3DGS set, which can be generated from an arbitrary number of views.
>
> ###  **Question 1. Video Comparisons**:
> We direct the reviewer to the original supplementary materials, which include qualitative video comparisons (file name: compare_with_major-baselines.mp4) of our method against the major baselines.
>
> To enhance the discussion and facilitate a more comprehensive visual assessment, we provide another video (file name:_compare_with_diffusion-sparse-baselines.mp4_) comparing our method against the diffusion-based and sparse-view baselines (Weakness 1/3 and Weakness 3/3).

---

> > ### Comment · Reviewer_K5mf · 2024-11-26
> > **thanks for the responses**
> >
> > Thank you for the detailed responses from the authors. The topic of out-of-distribution (OOD) synthesis is undoubtedly important and has many applications. Among these, object-level surface reconstruction for few-shot asset creation and novel view synthesis of large-scale scenes are particularly critical.
> >
> > I understand that diffusion model-based methods might produce inconsistencies when generating new images. However, these methods can generate views completely unseen during training, a limitation often encountered with transformer-based feed-forward methods.
> >
> > Additionally, the reviewer strongly recommends that the authors focus on object-level surface reconstruction (e.g., using 2D-GS on DTU datasets) and scene-wise reconstruction for novel view synthesis under a few-shot setting (e.g., employing both generalizable and feed-forward neural renderers on MipNeRF360 datasets).
> >
> > Incorporating these discussions and experiments will significantly enhance the strength of the claims and contributions within the paper.

---

> > > ### Author Response · Authors · 2024-11-26
> > > **Discussion about 'Hallucinating unseen views'**
> > >
> > > We sincerely thank the reviewer for their thoughtful response and valuable insights. We fully agree with the reviewer’s emphasis on the benefits of generative priors, as well as the significance of object-level surface reconstruction and few-shot scene-wise reconstruction. To address these points in greater detail, we are pleased to provide the following discussion.
> > >
> > > ### **1. Hallucinating unseen views**
> > > We agree that diffusion-based methods excel at inpainting or inventing unseen content—an impressive capability not easily matched by transformer-based feed-forward models. This feature is particularly valuable for creative applications such as AI-assisted design and 3D asset creation. We firmly believe that any work aimed at AI-driven creative tasks should benefit significantly from leveraging diffusion-based models [6,7].
> > >
> > > However, beyond AI creation, there exist real-world scenarios where users can provide a relatively large number of input views to disambiguate scene information, and novel view synthesis (NVS) techniques must prioritize accurate and faithful renderings of the true scene without introducing hallucinated content. For example, in surgical digitalization [20], surgeons require precise and accurate visualizations of incisions; even minor hallucination errors by the system could lead to severe, potentially fatal consequences. Similarly, in a traffic monitoring system that visualizes a bird’s-eye view of the city street, hallucinating non-existent pedestrians or vehicles could result in misinformation and critical decision-making errors.
> > >
> > > We have conducted extensive comparisons with diffusion-based methods in both our original work and the rebuttal, exploring various strategies to further improve their performance in the OOD-NVS setup. These efforts included fine-tuning the methods using our curated dataset and addressing inconsistencies by distilling a 3D geometry structure (3DGS). Our method consistently outperforms these approaches in all variations, both quantitatively and qualitatively:
> > >
> > > | Results on GSO-OOD | PSNR  | SSIM  | LPIPS   |
> > > |---|---|---|---|
> > > | SyncDreamer [1] (finetune) | 11.86 | 0.518  | 0.451 |
> > > | SV3D [2] (0-shot) | 10.93  | 0.498 | 0.455 |
> > > | SV3D$\rightarrow$3DGS | 14.19 | 0.562  | 0.405 |
> > > |  EscherNet [3] (0-shot) |  13.74 | 0.585  | 0.367|
> > > |  EscherNet (finetune) | 16.57 | 0.633  | 0.273 |
> > > |  EscherNet$\rightarrow$3DGS | 18.88 | 0.701  | 0.258 |
> > > | 3DGS [8] |   21.78  | 0.746  | 0.250 |
> > > | Ours |   **25.01**  | **0.863**  | **0.148** |
> > >
> > > We share the reviewer’s interest in diffusion models and fully acknowledge their exceptional ability to generate photorealistic results. However, as no existing work effectively addresses the hallucination issue of these models in our target scenarios, we aim to tackle this challenge in future research.

---

> > > ### Author Response · Authors · 2024-11-26
> > > **Discussion about 'object-level surface reconstruction' and 'few-shot scene-wise reconstruction'**
> > >
> > > ### **2. Object-level surface reconstruction**
> > > We agree that object-level surface reconstruction is a crucial task. Although our method is not specifically designed for surface reconstruction and extraction, as its primary focus is novel view synthesis, it is nonetheless capable of improving and refining the geometry of the input 3DGS set. To demonstrate this, we evaluate the mean absolute error (MAE) between the ground-truth depth and normal maps and their corresponding rendered depth and normal maps under out-of-distribution (OOD) views for 3DGS [8] and our method.
> > > Specifically, the depth maps for 3DGS and our method are rendered as the weighted average depth of Gaussian primitives, a standard approach for deriving depth maps from 3DGS, as implemented in the gsplat toolbox [9] and other 3DGS-related works [10,16]. We then compute the normal maps following 2DGS [17], using finite differences on the estimated surface derived from the depth maps.
> > > The results demonstrate that, in addition to enhancing rendering quality, our method significantly improves the accuracy of the rendered depth and normal maps:
> > >
> > > | Results on Objaverse-OOD | Depth-MAE (x1e-4) | Normal-MAE |
> > > |---|---|---|
> > > |3DGS [8] |6.70|0.239|
> > > |SplatFormer|**4.05**|**0.214**|
> > >
> > > We also include a visualization of the rendered depth map in Figure G.3 of the updated appendix.
> > >
> > > We acknowledge that there is still considerable room for improvement in surface reconstruction, particularly due to our use of 3DGS as the basis for 3D representation. Since 2DGS [17] produces more regularized depth and normal maps than 3DGS, our method could potentially yield better geometry results when applied to 2DGS refinement, as discussed in Lines 1128–1133 of the Appendix. Additionally, in our response to reviewer MA5d (Weakness: Additional Comparison with LaRa on a 4-view Setup), we trained a SplatFormer to refine LaRa’s 2DGS predictions using four input views, achieving noticeable improvements on the two OOD test sets:
> > >
> > >
> > > |            Four input views               | **Objaverse-OOD** | **GSO-OOD**  |
> > > |---------------------|------------------------------|------------------------------|
> > > |          | PSNR / SSIM / LPIPS | PSNR / SSIM / LPIPS |
> > > | LaRa               | 16.87 / 0.640 / 0.352       | 17.91 / 0.677 / 0.339       |
> > > | LaRa + SplatFormer | **18.29 / 0.688 / 0.275**       | **18.83 / 0.714 / 0.279**       |
> > >
> > > This result provides evidence that our proposed network and framework can also be integrated to refine 2DGS representations. While the focus of this paper is not on accurate surface reconstruction, we plan to incorporate 2DGS into our method and evaluate more detailed geometry results on related datasets, such as DTU, in _future work_.
> > >
> > > ### **3. Few-shot Scene-wise Reconstruction**
> > >
> > > We sincerely thank the reviewer for emphasizing the importance of few-shot scene-wise reconstruction. As discussed earlier, while we agree that few-shot settings are a critical problem, our focus is on a different but equally meaningful setup. In our approach, we assume that a relatively dense capture is available, albeit from a limited range of viewing angles, and a novel view from extreme angles is required.
> > >
> > > To experiment with this setup in complex scenes, we explore SplatFormer’s potential for real-world unbounded scenarios, as discussed in Section F: Limitations and Future Directions (Lines 1162–1185) of the originally submitted Appendix. To evaluate its performance, we applied our proposed framework to the MVImgNet dataset [5]. On the test set, our method outperforms 3DGS:
> > >
> > > | Method |PSNR |SSIM|LPIPS|
> > > |---|---|---|---|
> > > |3DGS [8] |19.81|0.728|0.432|
> > > |SplatFormer|**21.68**|**0.757**|**0.424**|
> > >
> > > We present the visual comparison in Figure F.1 and Figure G.2, which demonstrate that SplatFormer reduces floater artifacts and improves geometry in many cases. Future improvements could involve designing a novel multi-scale hierarchical point transformer architecture to handle larger scenes, as well as incorporating real-world training data alongside synthetic data. Additionally, since SplatFormer shows strong generalization on real-world objects (Table 2, Figure 5, and Figure F.5), it may be feasible to decompose the scene and process individual objects separately.
> > >
> > > Unbounded scene reconstruction from limited observations remains a challenging problem. Many prior-enhanced NVS methods [1,2,3,4] also focus on object-centric scenes. Methods like MVSplat [18] and PixelSplat [19], which use generalizable feed-forward neural renderers on MipNeRF360 datasets, focus on interpolating novel views between two input views.
> > >
> > > While our method excels in the object-centric settings discussed in this paper, it also shows promising potential for unbounded scenes. Improving the training strategy and network architecture will be a key focus in our future work.

---

> > > ### Author Response · Authors · 2024-11-26
> > >
> > > In summary, we greatly appreciate the reviewer’s suggestions and feedback. We have conducted extensive experiments to highlight the limitations of diffusion-based models, demonstrate improvements in geometry, and showcase our method's potential in unbounded scenes. In future work, we plan to further explore these directions and incorporate additional recommendations from the reviewer that extend beyond the scope of the current submission.
> > >
> > > References
> > >
> > > [18] Chen, Yuedong, et al. "Mvsplat: Efficient 3d gaussian splatting from sparse multi-view images." European Conference on Computer Vision. Springer, Cham, 2025.
> > >
> > > [19] Charatan, David, et al. "pixelsplat: 3d gaussian splats from image pairs for scalable generalizable 3d reconstruction." Proceedings of the IEEE/CVF Conference on Computer Vision and Pattern Recognition. 2024.
> > >
> > > [20] Hein, Jonas, et al. "Creating a Digital Twin of Spinal Surgery: A Proof of Concept." Proceedings of the IEEE/CVF Conference on Computer Vision and Pattern Recognition. 2024.

---

> > > > ### Comment · Reviewer_K5mf · 2024-11-27
> > > > **Response to authors**
> > > >
> > > > Thank you for the authors' comprehensive response, which addresses many of the concerns raised. I appreciate that the issues identified by various reviewers may not overlap completely. I recommend incorporating **more** visualizations of geometry (such as depth or meshes) for object-level reconstruction using 3D-GS.
> > > >
> > > > Additionally, testing the rendering quality on out-of-distribution scenes from different datasets, like **MipNeRF360** or **Tank and Temples** datasets, would greatly enhance our understanding of its generalization capabilities.

---

> > > > > ### Author Response · Authors · 2024-11-29
> > > > >
> > > > > We would like to thank the reviewer for their valuable suggestions.
> > > > >
> > > > > In addition to the depth visualization page in our manuscript (Figure G.3), we have provided additional normal visualization [here](https://1drv.ms/b/s!AsMtUYcbeDptbHOwORrU6XquDsE?e=8dosYR).  We also measured the mean absolute error (MAE) between the ground-truth depth and normal maps, and the corresponding rendered depth and normal maps, under out-of-distribution (OOD) views for 3DGS [8] and our method as below.
> > > > >
> > > > > | Results on Objaverse-OOD | Depth-MAE (x1e-4)$\downarrow$ | Normal-MAE$\downarrow$ |
> > > > > |---|---|---|
> > > > > |3DGS [8] |6.70|0.239|
> > > > > |SplatFormer|**4.05**|**0.214**|
> > > > >
> > > > > As pointed out in several relevant geometry enhanced 3DGS papers, extracting high-quality surface or meshes from 3DGS is intrinsically challenging due to its unstructured, explicit and discontinuous point-based nature (Sugar [18], GoF [19], PGSR [20]) and multiview inconsistency (2DGS [17]). Though constrained by the inherent limitations of 3DGS in exporting high-fidelity surfaces, our method still improves the accuracy of rendered depth map and normal map from 3DGS both quantitatively and qualitatively. Although accurate surface reconstruction is not the primary focus of this paper, we plan to integrate 2DGS or other geometrically accurate 3DGS representations into our framework in future work. This integration will enable us to evaluate more detailed geometric results on benchmark datasets, such as DTU.
> > > > >
> > > > > This work focuses on improving out-of-distribution views in object-centric scenes. While we acknowledge the limitations of our approach in unbounded scenes, preliminary experiments on MVImgnet (Figure F.1, Table F.1, and Figure G.2) demonstrate the potential of our framework for scene-wise reconstruction. An enhanced network architecture, along with a larger training dataset incorporating real-world scenes, could improve the model's generalization to more diverse scenes and capture setups. However, further improvements for unbounded scenes are outside the scope of this work, and including extensive experiments on this topic would not conform with ICLR's discussion rules and guidelines. We look forward to exploring this in future work.
> > > > >
> > > > > We sincerely thank the reviewer for their feedback and appreciate the insightful discussions.
> > > > >
> > > > > [17] Huang, Binbin, et al. "2d gaussian splatting for geometrically accurate radiance fields." ACM SIGGRAPH 2024 Conference Papers. 2024.
> > > > >
> > > > > [18] Guédon, Antoine, and Vincent Lepetit. "Sugar: Surface-aligned gaussian splatting for efficient 3d mesh reconstruction and high-quality mesh rendering." Proceedings of the IEEE/CVF Conference on Computer Vision and Pattern Recognition. 2024.
> > > > >
> > > > > [19] Yu, Zehao, Torsten Sattler, and Andreas Geiger. "Gaussian opacity fields: Efficient and compact surface reconstruction in unbounded scenes." arXiv preprint arXiv:2404.10772 (2024).
> > > > >
> > > > > [20] Chen, Danpeng, et al. "PGSR: Planar-based Gaussian Splatting for Efficient and High-Fidelity Surface Reconstruction." arXiv preprint arXiv:2406.06521 (2024).

---

> > > > > > ### Comment · Reviewer_K5mf · 2024-12-01
> > > > > > **Response from reviewer**
> > > > > >
> > > > > > Thank you for your detailed response. I appreciate the efforts to enhance the depth and normal visualizations and to discuss the challenges related to surface quality in 3DGS systems.
> > > > > >
> > > > > > Regarding the focus of your work, I believe that including **"Object"** in the title and abstract would more accurately reflect the current scope of the work, particularly as it relates to object-centric scenes. If you are willing to make this adjustment, I would be more inclined to **raise my score** for your submission, as it would better align the paper's content with its **title and abstract**, enhancing clarity for readers.

---

> > > > > > > ### Author Response · Authors · 2024-12-02
> > > > > > >
> > > > > > > Thank you for your thoughtful suggestion to include “object” in the title. We appreciate your effort to ensure our work is clearly and effectively presented.
> > > > > > >
> > > > > > > We carefully considered your feedback but decided to retain the current title, SplatFormer: Point Transformer for Robust 3D Gaussian Splatting. This choice reflects the broader scope of our contributions, which extend beyond object-centric applications to include experiments on unbounded scenes. Adding “object” to the title might inadvertently narrow the perceived applicability of our method.
> > > > > > >
> > > > > > > However, we recognize the importance of emphasizing object-centric aspects of our work. To address this, we will clarify these contributions explicitly in the abstract and introduction. These sections will highlight how our method enhances object representations in 3D Gaussian splatting, ensuring this focus is evident to readers.
> > > > > > >
> > > > > > > We believe this approach balances specificity with the broad applicability of our method while addressing your concern. We appreciate your insight and remain open to further suggestions to improve the clarity and impact of our submission.
> > > > > > > Thank you again for your valuable feedback.

---

### Author Response · Authors · 2024-11-25
**Summary of the authors' responses**

We sincerely thank all reviewers for their valuable comments and constructive suggestions. To facilitate discussion among the reviewers and the area chair, we have summarized the reviewers' feedback in the table below.

|Strengths|R-K5mf|R-MA5d|R-MWLJ| R-aVEM|
|----------|---|---|---|---|
|Addresses a significant research gap | &#10004;  | &#10004; | &#10004;  | &#10004; |
|Proposes a novel and sound method | &#10004;  |  |  | &#10004; |
|Construct novel OOD datasets | | &#10004;   |  | |
|Provides extensive experiments |  | &#10004; | &#10004;   | &#10004; |
|Achieves superior performance| &#10004;  | &#10004; | &#10004;  | &#10004; |
|The paper is well-written|  |  | | &#10004; |
||||||
|**Weaknesses** | | | |  |
|No diffusion priors | ✘|  |  |  |
|Unknown potential in unbounded scenes |✘ |  |  ✘| |
|Insufficient analysis on sparse-view baselines | ✘ | ✘ |  |  |
|Lack of geometry results|  |  | ✘ | |
||||||
|**Rating**| **5**|**6**|**6**|**8**|


We appreciate the reviewers' acknowledgment that our work addresses a significant research gap and their recognition of several strengths, including a novel dataset, extensive experiments, and a technically sound, high-performance method.

To address the reviewers' comments on limitations, we provided additional results and clarifications as follows:
1. **Comparison with Diffusion-based and Sparse-view Methods** (R-K5mf, R-MA5d): In addition to the existing comparisons with SyncDreamer [1], SSDNeRF [12], DiffBIR [13], and LaRa [4] in our original manuscript, we have included more numerical and visual comparisons with state-of-the-art open-source diffusion-based methods [1, 2, 3] and the sparse-view baseline LaRa [4]. We have also analyzed their limitations in the OOD-NVS setup.
2. **Potential in Unbounded Scenes** (R-K5mf, R-MWLJ): Building on the discussion in the original manuscript, supported by experimental results on MVImgNet [5] (Appendix F), we have expanded it with additional details, considerations for future improvements, and qualitative comparisons.
3. **Geometry Evaluation** (R-MWLJ): We have included qualitative and quantitative evaluations of depth and normal errors.

Furthermore, we addressed additional questions raised by the reviewers:
1. **Video Comparison** (R-K5mf): We directed the reviewer to the video included in our originally submitted supplementary material (_compare_with_major-baselines.mp4_), which provides visual comparison between our method and representative baselines. Additionally, we included a new video (compare_with_diffusion-sparse-baselines.mp4) comparing our method to [1, 2, 3, 4], highlighting the limitations of diffusion-based and sparse-view baselines in terms of hallucination errors and 3D inconsistencies.
2. **Computational Efficiency** (R-MA5d): We evaluated the inference time and memory usage of our model to demonstrate its efficiency.
3. **Training Setup** (R-MA5d):  We clarified the ratio of OOD views used during SplatFormer's training and included an ablation study on its effect.
4. **OOD Views Different from Training** (R-aVEM): We demonstrated that our method consistently improves novel test views across diverse viewing angles and distances and discussed its performance under varying input-view trajectories.

### **Additional Updates**:
* Additional Image Comparisons:  These have been added to Appendix G in the updated manuscript, including:
	* **Figure G.1**: Comparisons with diffusion-based and sparse-view baselines.
	* **Figure G.2**: Additional results in unbounded scenes.
	* **Figure G.3**: Geometry comparisons.
* Video Comparisons: Two videos are now available in the supplementary material:
	* **_compare_with_major-baselines.mp4_**: Originally submitted as part of our supplementary material, this video demonstrates comparisons between our method and major baselines.
	* **_compare_with_diffusion-sparse-baselines.mp4_**: Newly added to address the questions raised by R1-K5mf and R2-MA5d, this video compares our method with diffusion-based and sparse-view baselines.

We sincerely thank all reviewers and the area chair for their time, patience, and thoughtful feedback.

---

### Public Comment · ~Yiping_Ji1 · 2024-11-28
**Question about speed**

Hi Authors,
Thanks for this great work and I am quite interested in it. I was just wondering if there is extra cost in inference speed using splatformer compared with other methods.

---

### Meta-Review · Area_Chair_xf5h · 2024-12-17

**Metareview:**

This paper receives unanimous positive ratings of 6,8,8,8. The AC follows the recommendations of the reviewers to accept the paper. The reviewers comment that the method introduced by the paper is novel and the task is an important direction for rendering unseen test views which addressed a significant gap in the current research.

**Additional Comments On Reviewer Discussion:**

The reviewers asked for additional experiments and some further clarifications, which the authors managed to address well in the rebuttal and discussion phases.

---

### Decision · Program_Chairs · 2025-01-22

Accept (Spotlight)